# ANALYZING THE EXPRESSIVE POWER OF GRAPH NEURAL NETWORKS IN A SPECTRAL PERSPECTIVE

**Muhammet Balcilar,**[*] **Guillaume Renton, Pierre Héroux,**
**Benoit Gaüzère, Sébastien Adam, Paul Honeine**
Normandy University, LITIS Lab, University of Rouen Normandy, INSA Rouen Normandie
Rouen, 76000, France

## ABSTRACT

In the recent literature of Graph Neural Networks (GNN), the expressive power of models has been studied through their capability to distinguish if two given graphs are isomorphic or not. Since the graph isomorphism problem is NP-intermediate, and Weisfeiler-Lehman (WL) test can give sufficient but not enough evidence in polynomial time, the theoretical power of GNNs is usually evaluated by the equivalence of WL-test order, followed by an empirical analysis of the models on some reference inductive and transductive datasets. However, such analysis does not account the signal processing pipeline, whose capability is generally evaluated in the spectral domain. In this paper, we argue that a spectral analysis of GNNs behavior can provide a complementary point of view to go one step further in the understanding of GNNs. By bridging the gap between the spectral and spatial design of graph convolutions, we theoretically demonstrate some equivalence of the graph convolution process regardless it is designed in the spatial or the spectral domain. Using this connection, we managed to re-formulate most of the state-of-the-art graph neural networks into one common framework. This general framework allows to lead a spectral analysis of the most popular GNNs, explaining their performance and showing their limits according to spectral point of view. Our theoretical spectral analysis is confirmed by experiments on various graph databases. Furthermore, we demonstrate the necessity of high and/or band-pass filters on a graph dataset, while the majority of GNN is limited to only low-pass and inevitably it fails.
Code available at https://github.com/balcilar/gnn-spectral-expressive-power.

## 1 INTRODUCTION

Over the last five years, many Graph Neural Networks (GNNs) have been proposed in the literature of geometric deep learning (Veličković et al., 2018; Gilmer et al., 2017; Bronstein et al., 2017; Battaglia et al., 2018), in order to generalize the very efficient deep learning paradigm into the world of graphs. This large number of contributions explains a new challenge recently tackled by the community, which consists in assessing the expressive power of GNNs.

In this area of research, there is a consensus to evaluate the theoretic expressive power of GNNs according to equivalence of Weisfeiler-Lehman (WL) test order (Morris et al., 2019; Xu et al., 2019; Maron et al., 2019b;a). Hence, GNNs models are frequently classified as "as powerful as 1-WL", "as powerful as 2-WL", ..., "as powerful as k-WL". However, this perspective cannot make differences between two methods if they are as powerful as the same WL test order. Moreover, it does not always explain success or failure of any GNN on common benchmark datasets.

In this paper, we claim that analyzing theoretically and experimentally GNNs with a spectral point of view can bring a new perspective on their expressive power.

So far, GNNs have been generally studied separately as spectral based or as spatial based (Wu et al., 2019b; Chami et al., 2020). To the best of our knowledge, Message Passing Neural Networks (MPNNs) (Gilmer et al., 2017) and GraphNets (Battaglia et al., 2018) are the only attempts to merge

---

[*]muhammetbalcilar@gmail.com

both approaches in the same framework. However, these models are not able to generalize custom designed spectral filters, as well as the effect of each convolution support in a multi convolution case. The spatial-spectral connection is also mentioned indirectly in several cornerstone studies by Defferrard et al. (2016); Kipf & Welling (2017); Levie et al. (2019). Since the spectral-spatial interchangeability is missing, they did not propose to show spectral behavior of any graph convolution. Recent studies have also attempted to show, for a limited number of spatial GNNs, that they act as low-pass filters (NT & Maehara, 2019; Wu et al., 2019a). NT & Maehara (2019) concluded that using adjacency induces low-pass effects, while Wu et al. (2019a) studied a single spatial GNN's spectral behavior by assuming adding self-connection changes the given topology of the graph.

In this paper, we bridge the gap between spectral and spatial domains for GNNs. Our first contribution consists in demonstrating the equivalence of convolution processes regardless if they are defined as spatial or as spectral GNN. Using this connection, we propose a new general framework and taxonomy for GNNs as the second contribution. Taking advantage of this equivalence, our third contribution is to provide a spectral analysis of any GNN model. This spectral analysis is another perspective for the analysis of expressive power of GNNs. Our theoretical spectral analysis is confirmed by experiments on various well-known graph datasets. Furthermore, we show the necessity of high and/or band-pass filters in our experiments, while the majority of GNNs are limited to only low-pass filters and thus inevitably fail when dealing with these problems. The code used in this paper is available at https://github.com/balcilar/gnn-spectral-expressive-power.

The remainder of this paper is organized as follows. Section 2 introduces convolutional GNNs and presents existing approaches. In Section 3 and Section 4, we describe the main contributions mentioned above. Section 5 presents a series of experiments and results which validate our propositions. Finally, Section 6 concludes this paper.

## 2 PROBLEM STATEMENT AND STATE OF THE ART

Let $G$ be a graph with $n$ nodes and an arbitrary number of edges. Connectivity is given by the adjacency matrix $A \in \{0,1\}^{n \times n}$ and features are defined on nodes by $X \in \mathbb{R}^{n \times f_0}$, with $f_0$ the length of feature vectors. For any matrix $X$, we used $X_i$, $X_{:j}$ and $X_{i,j}$ to refer its $i$-th column vector, $j$-th row vector and scalar value on $(i,j)$ location, respectively. A graph Laplacian is $L = D - A$ (or $L = I - D^{-1/2}AD^{-1/2}$) where $D \in \mathbb{R}^{n \times n}$ is the diagonal degree matrix and $I$ is the identity. Through eigendecomposition, $L$ can be written by $L = U\text{diag}(\boldsymbol{\lambda})U^T$ where each column of $U \in \mathbb{R}^{n \times n}$ is an eigenvector of $L$, $\boldsymbol{\lambda} \in \mathbb{R}^n$ gathers the eigenvalues of L and diag(.) function creates a diagonal matrix whose diagonal elements are from a given vector. We use superscript to refer same kind variable as base. For instance, $H^{(l)} \in \mathbb{R}^{n \times f_l}$ refers node representation on layer $l$ whose feature dimension is $f_l$. A Graph Convolution layer takes the node representation of the previous layer $H^{(l-1)}$ as input and produces a new representation $H^{(l)}$, with $H^{(0)} = X$.

### 2.1 SPECTRAL APPROACHES

Spectral GNNs rely on the spectral graph theory (Chung, 1997). In this framework, signals on graphs are filtered using the eigendecomposition of graph Laplacian (Shuman et al., 2013). By transposing the convolution theorem to graphs, the spectral filtering in the frequency domain can be defined by $x_{flt} = U\text{diag}(\Phi(\boldsymbol{\lambda}))U^\top x$, where $\Phi(.)$ is the desired filter function. As a consequence, a graph convolution layer in spectral domain can be written by a sum of filtered signals followed by an activation function as in (Bruna et al., 2013), namely

$$H_j^{(l+1)} = \sigma\left(\sum_{i=1}^{f_l} U\text{diag}(F_i^{(l,j)})U^\top H_i^{(l)}\right), \qquad \text{for } j \in \{1, \ldots, f_{l+1}\}. \tag{1}$$

Here, $\sigma$ is the activation function, $F^{(l,j)} \in \mathbb{R}^{n \times f_l}$ is the corresponding weight vector to be tuned as used in (Henaff et al., 2015) for the single-graph problem known as non-parametric spectral GNN.

A first drawback is the necessity of Fourier and inverse Fourier transform by matrix multiplication of $U$ and $U^T$. Another drawback occurs when generalizing the approach to multi-graph learning problems. Indeed, the $k$-th element of the vector $F_i^{(l,j)}$ weights the contribution of the $k$-th eigenvector to the output. Those weights are not shareable between graphs of different sizes, which means a

different length of $F_i^{(l,j)}$ is needed. Moreover, even though the graphs have the same number of nodes, their eigenvalues will be different if their structures differ.

To overcome these issues, a few spatially-localized filters have been defined such as cubic B-spline (Bruna et al., 2013), polynomial and Chebyshev polynomial (Defferrard et al., 2016) and Cayley polynomial parameterization (Levie et al., 2019). With such approaches, trainable parameters are defined by $F_i^{(l,j)} = B\left[W_{i,j}^{(l,1)}, \ldots, W_{i,j}^{(l,s_e)}\right]^\top$, where each column in $B \in \mathbb{R}^{n \times s_e}$ is designed as a function of eigenvalues, namely $B_{k,s} = \Phi_s(\lambda_k)$, where $k = 1, \ldots, n$ denotes eigenvalue index, $s = 1, \ldots, s_e$ denotes index of filters and $s_e$ is the number of desired filters. Here, $W^{(l,s)} \in \mathbb{R}^{f_l \times f_{l+1}}$ is the trainable matrix for the $l$-th layer's $s$-th filter's.

## 2.2 SPATIAL APPROACHES

Spatial GNNs consider an $agg$ operator, which aggregates the neighborhood nodes, and an $upd$ operator, which updates the concerned node as follows:

$$H_{:v}^{(l+1)} = upd\Big(g_0(H_{:v}^{(l)}), agg\Big(g_1(H_{:u}^{(l)}) : u \in \mathcal{N}(v)\Big)\Big), \tag{2}$$

where $\mathcal{N}(v)$ is the set of neighborhood nodes and $g_0, g_1 : \mathbb{R}^{n \times f_l} \to \mathbb{R}^{n \times f_{l+1}}$ trainable models. The choice of $agg, upd, g_0, g_1$, and even $\mathcal{N}(v)$, determines the capability of model.

The vanilla GNN (known by GIN-0 in (Xu et al., 2019)) uses the same weights in $g_0$ and $g_1$. $\mathcal{N}(v)$ is the set of connected nodes to $v$, $agg$ is the sum of all connected node values and $upd(x, y) := \sigma(x + y)$ where $\sigma$ is an elementwise nonlinearity. GCN has the same selection but normalizes features as in (Kipf & Welling, 2017). Hamilton et al. (2017) used separated weights in $g_0$ and $g_1$, which means that two sets of trainable weights are applied on self feature and neighbor nodes. Other approaches defined multi neighborhood and used different $g_i$ for different kind of neighborhood. For instance, Duvenaud et al. (2015) defined the neighborhood according to node label and/or degree, Niepert et al. (2016) reordered the neighbor nodes and used the same model $g_i$ to neighbors according to their order.

These spatial GNNs use sum or normalized sum over $g_i$ in equation 2. Other methods weighted this summation by another trainable parameter, where the weights can be written by the function of node and/or edge features in order to make the convolutions more productive, such as graph attention networks (Veličković et al., 2018), MoNet (Monti et al., 2017), GatedGCN (Bresson & Laurent, 2018) and SplineCNN (Fey et al., 2018).

## 3 BRIDGING SPATIAL AND SPECTRAL GNNS

In this section, we define a general framework which includes most of the well-know GNN models, including euclidean convolution and models which use anisotropic update schema such as in Veličković et al. (2018); Bresson & Laurent (2018).

When $upd(x, y) = \sigma(x + y)$, $agg$ is a sum (or weighted sum) of the defined neighborhood nodes contributions and $g_i$ applies linear transformation, one can trivially show that mentioned spatial GNNs can be generalized as propagation of the node features to the neighboring nodes followed by feature transformation and activation function of the form

$$H^{(l+1)} = \sigma\Big(\sum_s C^{(s)} H^{(l)} W^{(l,s)}\Big), \tag{3}$$

where $C^{(s)} \in \mathbb{R}^{n \times n}$ is the $s$-th convolution support that defines how the node features are propagated to the neighboring nodes. Within this generalization, GNNs differ from each other by the choice of convolution supports $C^{(s)}$. This formulation generalizes many different kinds of Graph Convolutions, as well as Euclidean domain convolutions, which can be seen in Appendix A with the detailed schema.

**Definition 1.** *A **Trainable-support** is a Graph Convolution Support $C^{(s)}$ with at least one trainable parameter that can be tuned during training. If $C^{(s)}$ has no trainable parameters, i.e. when the supports are pre-designed, it is called a **fixed-support** graph convolution.*

In the trainable support case, supports can be different in each layer, which can be shown by $C^{(l,s)}$ for the $s$-th support in layer $l$. Formally, we can define a trainable support by:

$$\left(C^{(l,s)}\right)_{v,u} = h_{s,l}\left(H^{(l)}_{:v}, H^{(l)}_{:u}, E^{(l)}_{v,u}, A\right), \tag{4}$$

where $E^{(l)}_{v,u}$ shows edge features on layer $l$ from node $v$ to node $u$ if it is available and $h(.)$ is any trainable model parametrized by $(s,l)$.

**Theorem 1.** *Spectral GNN parameterized with $B$ of entries $B_{i,j} = \Phi_j(\lambda_i)$, defined as*

$$H^{(l+1)}_j = \sigma\left(\sum_{i=1}^{f_l} U \operatorname{diag}\left(B\left[W^{(l,1)}_{i,j}, \ldots, W^{(l,s_e)}_{i,j}\right]^{\top}\right)U^{\top}H^{(l)}_i\right), \tag{5}$$

*is a particular case of framework in equation 3 with the convolution kernel set to*

$$C^{(s)} = U \operatorname{diag}(\Phi_s(\boldsymbol{\lambda}))U^{\top}. \tag{6}$$

The proof can be found in Appendix B. This theorem is general and it covers many well-known spectral GNNs, such as non-parametric spectral graph convolution (Henaff et al., 2015), polynomial parameterization (Defferrard et al., 2016), cubic B-spline parameterization (Bruna et al., 2013), CayleyNet (Levie et al., 2019) and also any custom designed graph convolution. From Theorem 1, one can see that the spatial and spectral GNNs work all the same way. Therefore, Fourier calculations are not necessary when convolutions are parameterized by $B$. As a consequence of Theorem 1, one can see that the separation of spectral and spatial GNNs is just an interpretation. The only difference is the way convolution supports are designed: either in the spectral domain or in the spatial one.

**Definition 2.** *A **Spectral-designed** graph convolution refers to a convolution where supports are written as a function of eigenvalues ($\Phi_s(\boldsymbol{\lambda})$) and eigenvectors ($U$) of the corresponding graph Laplacian (equation 6). Thus, each convolution support $C^{(s)}$ has the same frequency response $\Phi_s(\boldsymbol{\lambda})$ over different graphs. Graph convolution out of this definition is called **spatial-designed** graph convolution.*

**Corollary 1.1.** ***The frequency profile** of any given graph convolution support $C^{(s)}$ can be defined in spectral domain by*

$$\Phi_s(\boldsymbol{\lambda}) = diag^{-1}(U^{\top}C^{(s)}U). \tag{7}$$

*where $diag^{-1}(.)$ returns the vector made of the diagonal elements from the given matrix.*

The proof of this corollary is given in Appendix C. This corollary leads to the spectral analysis of any given graph convolution support, including spatial-designed convolutions. Since the spatial-designed convolutions do not fit into equation 6, $U^{\top}C^{(s)}U$ is not a diagonal matrix. Therefore, we also compute the **full frequency profile** by $\Phi_s = U^{\top}C^{(s)}U$, which includes all eigenvectors pairwise contributions for spatial-designed convolutions.

## 4 THEORETICAL FREQUENCY RESPONSE OF GRAPH CONVOLUTIONS

This section aims at providing a theoretical understanding of the graph convolution process through an analysis in the spectral domain of existing GNNs. To the best of our knowledge, no one has led such an analysis concerning graph convolutions in the literature. This analysis is based on a reformulation of existing graph convolutions in our general framework (equation 3), and based on deriving analytical expressions of $\Phi_s(\boldsymbol{\lambda})$ (equation 7 in Corollary 1.1) for each convolution support of concerned graph convolution process. All proofs are provided in Appendices.

The theoretical frequency response of ChebNet (Defferrard et al., 2016) convolutions is given by the following theorem.

**Theorem 2.** *The theoretical frequency response of each support of ChebNet can be defined as*

$$\Phi_1(\boldsymbol{\lambda}) = \mathbf{1}, \quad \Phi_2(\boldsymbol{\lambda}) = \frac{2\boldsymbol{\lambda}}{\lambda_{\max}} - \mathbf{1}, \quad \Phi_k(\boldsymbol{\lambda}) = 2\Phi_2(\boldsymbol{\lambda})\Phi_{k-1}(\boldsymbol{\lambda}) - \Phi_{k-2}(\boldsymbol{\lambda}), \tag{8}$$

*where $\mathbf{1}$ is the vector of ones and $\lambda_{\max}$ is the maximum eigenvalue.*

The proof of Theorem 2 is given in Appendix D. Since it has no trainable parameter in the supports and all support frequency responses do not depend on the graph, we can classify ChebNet as *spectral-designed fixed-support* graph convolution.

The theoretical frequency response of CayleyNet (Levie et al., 2019) convolution is given in the following theorem, and its proof is given in Appendix E.

**Theorem 3.** *The theoretical frequency response of each support of CayleyNet can be defined as*

$$\Phi_s(\boldsymbol{\lambda}) = \begin{cases} \mathbf{1} & \text{if } s = 1 \\ \cos(\frac{s}{2}\theta(h\boldsymbol{\lambda})) & \text{if } s \in \{2, 4, \ldots, 2r\} \\ -\sin(\frac{s-1}{2}\theta(h\boldsymbol{\lambda})) & \text{if } s \in \{3, 5, \ldots, 2r+1\} \end{cases} \tag{9}$$

*where $h$ is a trainable scalar and $\theta(x) = atan2(-1, x) - atan2(1, x)$.*

Since it has a trainable parameter $h$ in the supports and all support frequency responses do not depend on the graph, we can classify CayleyNet as *spectral-designed trainable-support* graph convolution.

GCN (Kipf & Welling, 2017) uses a single convolution support and its theoretical frequency response is defined approximately in the following theorem, and its proof is given in Appendix F.

**Theorem 4.** *The theoretical frequency response of GCN support can be approximated as*

$$\Phi(\boldsymbol{\lambda}) \approx \mathbf{1} - \boldsymbol{\lambda}\overline{p}/(\overline{p}+1), \tag{10}$$

*where $\overline{p}$ is the average node degree in the graph.*

Since its support has no trainable parameter but the frequency response is not independent of the graph, we can classify GCN as *spatial-designed fixed-support* graph convolution.

Graph Isomorphism Network (GIN) defined in Xu et al. (2019) has attracted a lot of interests from the community, mostly because of its simple convolution mechanism. It has a single convolution support and its theoretical frequency response is given in the following theorem:

**Theorem 5.** *The theoretical frequency response of GIN support can be approximated as*

$$\Phi(\boldsymbol{\lambda}) \approx \overline{p}\left(\frac{1+\epsilon}{\overline{p}} + \mathbf{1} - \boldsymbol{\lambda}\right) \tag{11}$$

*where $\epsilon$ is a trainable scalar.*

The proof of this theorem is in Appendix G. Since its support has trainable parameters but the frequency response depends on the graph structure, we classify GIN as *spatial-designed trainable-support* graph convolution.

Graph attention networks (GATs) in (Veličković et al., 2018) proposes an application for graph world of the attention mechanism from Vaswani et al. (2017). Due to the fact that graphs are invariant to the node order, GAT cannot use positional encoding. In addition, instead of considering that all nodes are connected to each other, GAT just assigns attention weights to the node itself and the connected ones according to adjacency (sparse attention). Thus, we can see its convolution support as weighted, self loop added adjacency. GAT can be represented in our framework in equation 3 by defining trainable convolution supports as follows:

$$\left(C^{(l,s)}\right)_{v,u} = \frac{e_{v,u}}{\sum_{k \in \tilde{\mathcal{N}}(v)} e_{v,k}}, \tag{12}$$

where $e_{v,u} = \exp\left(\sigma(\mathbf{a}^{(l,s)}[H_{:v}^{(l)}W^{(l,s)}||H_{:u}^{(l)}W^{(l,s)}])\right)$, and $\mathbf{a}^{(l,s)}$ is another trainable weight. Convolution support will be calculated from node $v$ to each element of $\tilde{\mathcal{N}}(v)$, which shows the self-connection added neighborhood. Thus, we classify GAT as *spatial-designed trainable-support* graph neural network in our framework. Since convolution supports are function of connected node features, a theoretical frequency response is not possible to formulate. All studied models are summarised in Table 1.

## 5    EXPERIMENTAL RESULTS

This section is dedicated to empirical spectral analysis of existing GNNs on some certain graphs to validate the theoretical results and also performance analysis of these GNNs on a benchmark graph

Table 1: Summary of the studied GNN models.

| | Design | Support Type | Convolution Matrix | Frequency Response |
|---|---|---|---|---|
| MLP | Spectral | Fixed | $C = I$ | $\Phi(\boldsymbol{\lambda}) = \mathbf{1}$ |
| GCN | Spatial | Fixed | $C = \tilde{D}^{-0.5}\tilde{A}\tilde{D}^{-0.5}$ | $\Phi(\boldsymbol{\lambda}) \approx \mathbf{1} - \boldsymbol{\lambda}\overline{p}/(\overline{p}+1)$ |
| GIN | Spatial | Trainable | $C = A + (1+\epsilon)I$ | $\Phi(\boldsymbol{\lambda}) \approx \overline{p}\left(\frac{1+\epsilon}{\overline{p}} + \mathbf{1} - \boldsymbol{\lambda}\right)$ |
| GAT | Spatial | Trainable | $C_{v,u}^{(s)} = e_{v,u}/\sum_{k\in\tilde{\mathcal{N}}(v)} e_{v,k}$ | NA |
| CayleyNet[a] | Spectral | Trainable | $C^{(1)} = I$ 
 $C^{(2r)} = Re(\rho(hL)^r)$ 
 $C^{(2r+1)} = Re(\mathbf{i}\rho(hL)^r)$ | $\Phi_1(\boldsymbol{\lambda}) = \mathbf{1}$ 
 $\Phi_{2r}(\boldsymbol{\lambda}) = \cos(r\theta(h\boldsymbol{\lambda}))$ 
 $\Phi_{2r+1}(\boldsymbol{\lambda}) = -\sin(r\theta(h\boldsymbol{\lambda}))$ |
| ChebNet | Spectral | Fixed | $C^{(1)} = I$ 
 $C^{(2)} = 2L/\lambda_{\max} - I$ 
 $C^{(s)} = 2C^{(2)}C^{(s-1)} - C^{(s-2)}$ | $\Phi_1(\boldsymbol{\lambda}) = \mathbf{1}$ 
 $\Phi_2(\boldsymbol{\lambda}) = 2\boldsymbol{\lambda}/\lambda_{\max} - \mathbf{1}$ 
 $\Phi_s(\boldsymbol{\lambda}) = 2\Phi_2(\boldsymbol{\lambda})\Phi_{s-1}(\boldsymbol{\lambda}) - \Phi_{s-2}(\boldsymbol{\lambda})$ |

[a] $\rho(x) = (x - \mathbf{i}I)/(x + \mathbf{i}I)$

dataset to demonstrate the necessity of having various frequency responses convolution supports. The implementation and the introduced datasets are publicly available[1].

## 5.1 SPECTRAL ANALYSIS RESULTS

All empirical analyses are based on obtaining convolution supports matrix for certain GNN model, followed by equation 7 to obtain the frequency response. In our analysis, we used three graphs independently: the first is a 1D signal encoded as a regular circular line graph with 1001 nodes; the others are the well-known Cora and CiteSeer graphs with 2708 and 3327 nodes respectively (Yang et al., 2016). Besides, we used 2 different collections of graph datasets, ENZYMES and PROTEIN, which have 600 and 1113 graph respectively (Kersting et al., 2016). The details of the graphs can be found in Appendix L.

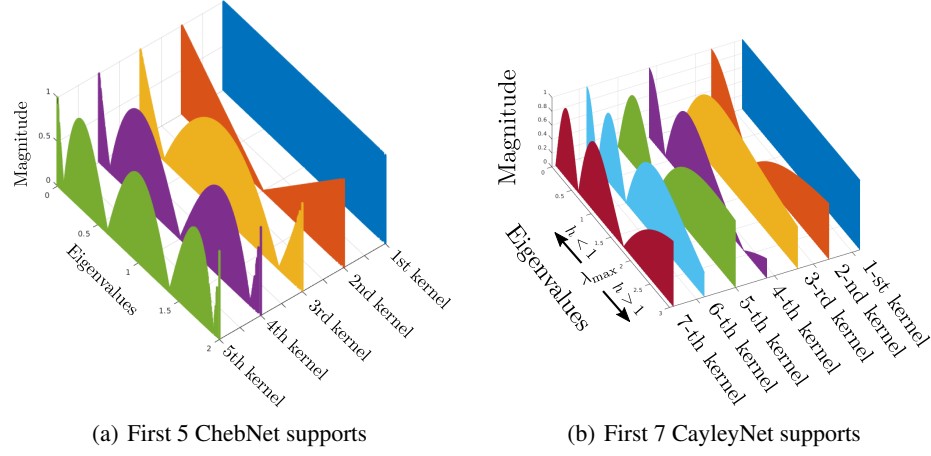

(a) First 5 ChebNet supports  (b) First 7 CayleyNet supports

Figure 1: Frequency profiles ($\Phi_s(\boldsymbol{\lambda})$)

Since ChebNet and CayleyNet are spectral-designed, their frequency responses do not change for different graphs. They are presented in Figure 1 for first 5 and 7 supports respectively. The results in Figure 1 confirm the theoretical analyses in Theorem 2 and Theorem 3. The full frequency profiles are not illustrated because they consist of zeros outside the diagonal. Analyzing the frequency profile of ChebNet, one can argue that the convolutions mostly cover the spectrum. However, none of the kernels focuses on some certain parts of the spectrum. As an example, the second kernel is mostly a low-pass and high-pass filter and stops the middle band, while the third one passes very high, very low and middle bands, but stops almost first and third quarter of the spectrum. Therefore, if the relation between input-output pairs can be figured out by just a low-pass, high-pass or some specific

---

[1]https://github.com/balcilar/gnn-spectral-expressive-power

band-pass filter, a high number of convolution kernels is needed. However, in the literature, only 2 or 3 kernels are generally used in experiments (Defferrard et al., 2016; Kipf & Welling, 2017).

The scale parameter $h$ in CayleyNet affects the x-axis scaling, but does not change the global shape. When $h = 1$, frequency profiles can be defined within the range $[0, 2]$ (because $\lambda_{\max} = 2$ in all three test graphs). If $h = 1.5$, the frequency profile can be defined till $1.5\lambda_{\max} = 3$ in Figure 1 and rescale axis label from $[0, 3]$ to $[0, 2]$ in original range. Learning the scaling of eigenvalues may seem advantageous. However, it induces extra computational cost in order to calculate the new convolution supports in every learning epoch. In addition, similarly to ChebNet, CayleyNet does not have any band specific convolutions, even when considering different scaling factors.

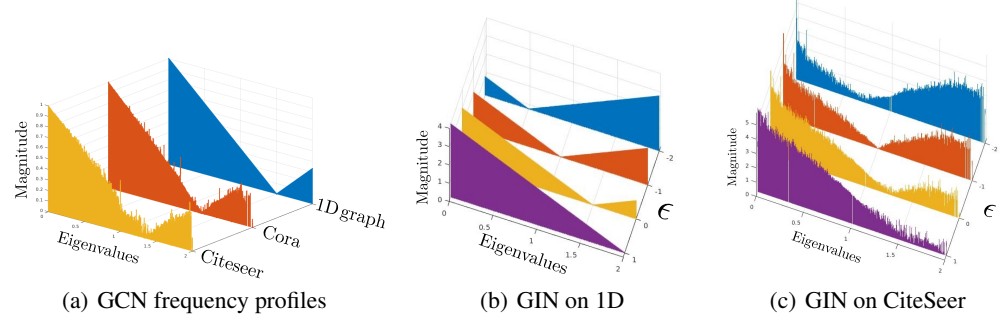

| (a) GCN frequency profiles | (b) GIN on 1D | (c) GIN on CiteSeer |

Figure 2: Frequency profiles of GCN on 1D, Cora, CiteSeer graph and GIN on 1D and CiteSeer graph with $\epsilon = 1, 0, -1, -2$

When the given graph is a regular graph where each node degree is the same (2 for 1D graph case), theoretical frequency responses become certain as seen in Figure 2a in blue for GCN and Figure 2b for GIN. When $\epsilon = 2$, 1D graph's ($\bar{p} = 2$) frequency responses of GCN and GIN are the same except scaling factor as seen in blue Figure 2a yellow in Figure 2b. However in realistic graphs, both GIN and GCN are not spectral-designed, their frequency responses differ for different graphs. As Theorem 4 and Theorem 5 demonstrate, GCN's and GIN's frequency responses depend on the average node degree. GCN's cut-off frequency decrease by increasing the $\bar{p}$ while $\bar{p}$ acts as scaling factor on GIN's frequency response. This analysis leads us to understand that GCN works as low-pass filter and does not cover the whole spectrum. This approach is not able to learn relations that can be represented by high-pass or band-pass filtering. Hence, even though it gives very good results on a single graph node classification problem in Kipf & Welling (2017), it may fail for problems where discriminant information lies in particular frequency bands. Therefore, such an approach can be considered as problem specific.

In order to create some variations between low-pass to high-pass, having trainable parameter in GIN's convolution support seems advantageous. But, since it is not spectral-designed, there is no guarantee that it has exactly the same spectral profiles for different graphs. Besides, its low-pass shape (where $\epsilon$ is high) is a linearly decreasing function, thus it is not a strong low-pass that generally natural graph problems need. Using more stacked layer may be a solution. In addition, this convolution cannot focus on some certain bands if the problem needs.

Since the GAT's convolution supports are function of connected nodes feature, frequency profiles cannot be directly computed similarly to previous ones. Thus, we proposed to obtain frequency response by two ways, one is the expected frequency responses among simulations, the other is the frequency responses of trained model for any specific graph learning problem.

We calculated the expected frequency responses of GAT convolution supports on Cora graph by simulation of randomly created 240 possible attention weights. The expected value of simulated support's frequency response and its standard deviation are shown in Figure 3a. This result gives an idea about the capability of the model on spectral domain, without being the true learned convolution support. In addition, the simulation is just for the first layer, because the first layer's input is known without learning. Besides, we also provide in Figure 3b-c the frequency responses of all learned GAT attention head's in all layers for all the graphs of ENZYMES and PROTEINS datasets respectively (in our model, there are two GNN layer consisting of 25 attention head). Since there is no significant

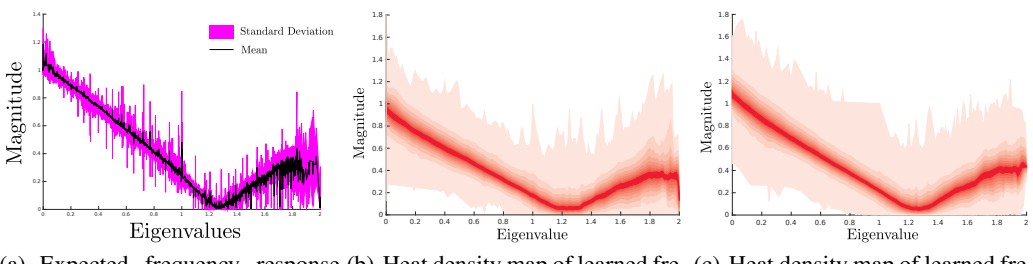

(a) Expected frequency response from Simulation on Cora

(b) Heat density map of learned frequency response on ENZYMES

(c) Heat density map of learned frequency response on PROTEINS

Figure 3: Frequency profiles of GAT

differences between frequency responses in different layer or different attention head, we stacked all together in the same heat map.

As one can see, the mean standard frequency profile has a similar shape than those of GCN and GIN-0 which are methods that use self-looped added (normalized or not) adjacency matrix as convolution support. Variations on the frequency profile induce more variations on output signal when compared to GCN and GIN-0. However, the variation on frequency profile might not be sufficient in problems that need some specific band-pass filters.

## 5.2 PERFORMANCE ANALYSIS OF GNNS

First, our goal was to assess empirically the ability of GNN models to produce the desired frequency effects on given graph signals. With the conducted experiments detailed in Appendix I, we can outline the empirical results as follows. GCN and GAT can produce low-pass effects but not band-pass or high-pass, while GAT has better variation on frequency profile; these empirical results corroborate the theoretical analysis of this paper, namely Theorem 4 and Section 5.1. Thanks to its trainable parameter $\epsilon$, GIN can do better on producing low-pass and high-pass effects, but not band-pass, as demonstrated by Theorem 5. However, the spectral-designed ChebNet always outperformed the rest with a huge margin, which is not surprising.

Secondly, we measured the generalization capability of the GNN model for graph classification task as a toy example where the graph classes depend on the frequency of the signal on the graph. The conducted experiments are described in Appendix J. Again, we seen that GCN and GAT performed worse than GIN, because of their inability to catch necessary frequency components. Besides, thanks to the spectral-designed convolutions, ChebNet can catch the underlying patterns on the graphs and finally achieves better results.

Some graph problems naturally just need low-pass filtering, as we argued in Appendix K. Having spectral ability may increase the complexity of the model, which may result in a negative effect. However, some other problems might need various kind of filters, like image understanding problems. In our last experiment, we use the superpixel version of MNIST dataset (MNIST-75) [2] to show an example of graph problems that need various filtering. In MNIST-75, images are segmented into around 75 regions by the SLIC superpixel segmentation algorithm (Achanta et al., 2012). Regions constitute the nodes of the graph and edges correspond to connection between regions in the image. The average pixel value of this region was assign to node, giving one continuous value. The dataset also includes the center position of each region, but we excluded that information to make the problem more realistic and harder in terms of graph research. The dataset consists of 55K graphs for training, 5K graphs for validation and 10K for testing. Details and some illustrations of the dataset can be found in Appendix L.

We use 3 hidden graph convolution layers that have 64, 128, and 128 features respectively, followed by a global mean operator as graph readout layer, and ended by a fully connected layer with 10 outputs corresponding to the number of classes. To understand the effect of graph convolution, we apply the tests on 3 different inputs: the first one uses node degree as feature, the second one uses

---

[2]https://graphics.cs.tu-dortmund.de/fileadmin/ls7-www/misc/cvpr/mnist-superpixels.tar.gz

Table 2: Test set accuracies on MNIST superpixel dataset

| Node feature | MLP | GCN | GIN | GAT | CayleyNet | ChebNet |
|---|---|---|---|---|---|---|
| Node degree | 11.29±0.5 | 15.81±0.8 | 32.45±1.2 | 31.72±1.5 | 45.61±1.7 | 46.23±1.8 |
| Pixel value | 12.11±0.5 | 11.35±1.1 | 64.96±3.9 | 62.61±2.9 | 88.41±2.1 | 91.10±1.9 |
| Both | 25.10±1.2 | 52.98±3.1 | 75.23±4.1 | 82.73±2.1 | 90.31±2.3 | 92.08±2.2 |

pixel values and the last one uses both information. Implementation details and hyperparameter tuning can be found in Appendix L.

Table 2 gives the mean and standard deviation of the accuracy obtained over 10 runs on the test set, with different seed numbers. It is well known that the image version of the MNIST dataset can be processed by any ordinary CNN architecture, which is able to apply various filtering operations. Hence, we argue that superpixel graph of MNIST is a good candidate to show if the graph data needs various kind of filtering. As seen in Table 2, MLP and GCN cannot do significantly better than a random classifier when using only node degree or pixel value as input. That means that the distribution of node degrees or pixel values has no significant meaning for classification. When both node degree and pixel values are given, the accuracy of GCN is increased, but remains behind the best results. GIN and GAT outperform GCN in each case, but their performances remain behind those of ChebNet and CayleyNet, which are spectral-designed with supports that cover the spectrum.

## 6 FINAL REMARKS

In this paper, we have shown that most influential graph convolutions such as (Kipf & Welling, 2017; Veličković et al., 2018) operate as low-pass filters and some have a very limited ability on producing high-pass in addition to low-pass filtering effect such as (Xu et al., 2019).

Interestingly, while being restricted to low-pass filters, they obtain state-of-the-art performance on reference node classification problems such as Cora, CiteSeer and Pubmed (Yang et al., 2016). These good results on these particular problems are induced by the nature of the graphs to be processed. Indeed, citation network problems, which are heavily assortative, are inherently low-pass filtering problems.

It is worth noting that, if we use enough convolution kernels, the frequency response of ChebNet kernels (Defferrard et al., 2016; Levie et al., 2019) covers nearly all frequency profiles. However, these frequency responses are not specific to special bands of frequency. It means that they can act as high-pass filters, but not as Gabor-like special band-pass filters,if a low number of convolution supports are used (e.g. 3). Getting any arbitrary band-pass effect requires a large number of convolution kernels, which makes the convolution not spatially-localized and increases the computational complexity.

As a conclusion, we claim that graph convolutions are problem specific and not problem agnostic. To have problem agnostic solutions, graph convolutions need to be able to produce necessary or at least plenty of different frequencies in output signal profile. The frequency profile of graph convolutions is not the single issue to be taken into account. But it is definitely one of the important perspectives that we need to pay attention. We point out that using only low-pass GNNs may not be a good choice for many graph problems. Finally, the convolution design can be considered as the tuning of hyperparameters or it can be automatically designed by another secondary unsupervised task with respect to the problem domain. Our future work will investigate this track. Experiments conducted in Section 5 provided empirical results to validate the theoretical analysis conducted in this paper.

ACKNOWLEDGMENTS

This work was partially supported by the French Agence National de Recherche (ANR), grant APi (ANR-18-CE23-0014), and the PAUSE Program (Collège de France).

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

## A    GENERALISATION OF FRAMEWORK

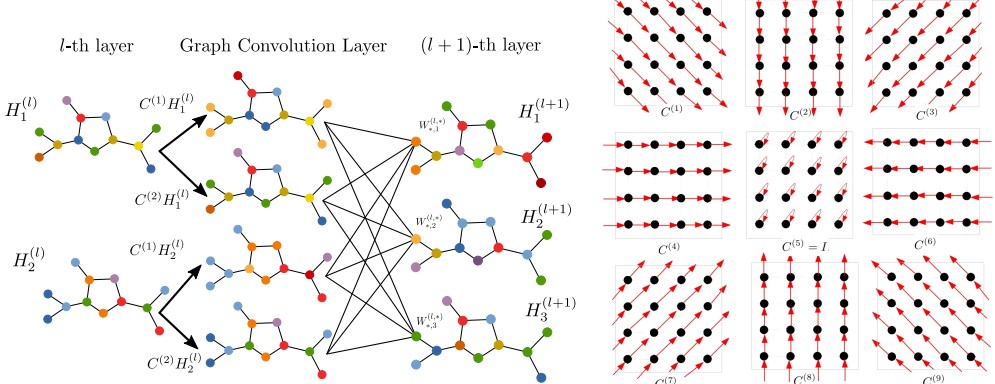

Figure 4: **a)** Schematic of the graph convolution layer defined in equation 3. The graph has 12 nodes and 12 edges. In the $l$-th layer, each node has a 2-length feature vector $H_1^{(l)}$ and $H_2^{(l)}$ represented by colors. The $l + 1$-th layer, it has a 3-length feature vector, denoted $H_1^{(l+1)}$, $H_2^{(l+1)}$ and $H_3^{(l+1)}$. Two convolution supports $C^{(1)}$ and $C^{(2)}$ are used. This architecture has 12 trainable parameters, omitting biases if the convolution supports $C^{(1)}$ and $C^{(2)}$ are fixed. **b)** Graph convolution supports for 2D Euclidean domain signal to perform convolution process by $3 \times 3$ mask.

Our selection of GNN generalization in equation 3 can be shown in Figure 4a with a detailed schematic of graph convolution layer on a sample graph signal. This framework can also generalize the Euclidean domain convolution layer. For 2D signal convolution by $\widetilde{W}^{(l)} \in \mathbb{R}^{3 \times 3}$ mask, we can define 9 different convolution supports denoted by $C^{(1)} \ldots C^{(9)} \in \{0, 1\}^{16 \times 16}$ in Figure 4 for sample signal (e.g. image) shown by $\widetilde{H}^{(l)} \in \mathbb{R}^{4 \times 4}$. When we stack all node values (e.g. pixels) into column vector $H^{(l)} \in \mathbb{R}^{16 \times 1}$, and $W^{(l,s)}$ shows $s$-th scalar weight in stacked $\widetilde{W}^{(l)}$, we can write equivalence of Euclidean convolution by:

$$\widetilde{H}^{(l)} \circledast \widetilde{W}^{(l)} = \sum_{s=1}^{9} C^{(s)} H^{(l)} W^{(l,s)}, \tag{13}$$

One can see that in Euclidean domain, the supports can be designed by relative position of the nodes which is not the case in graph world.

## B    PROOF OF THEOREM 1

*Proof.* First, let us expand the $B$ matrix by introducing its columns denoted $\Phi_1(\boldsymbol{\lambda}), \ldots, \Phi_S(\boldsymbol{\lambda}) \in \mathbb{R}^n$:

$$H_j^{(l+1)} = \sigma \left( \sum_{i=1}^{f_l} U \text{diag} \Big( \sum_{s=1}^{S} W_{i,j}^{(l,s)} \Phi_s(\boldsymbol{\lambda}) \Big) U^\top H_i^{(l)} \right). \tag{14}$$

Now, we distribute $U$ and $U^\top$ over the inner summation:

$$H_j^{(l+1)} = \sigma \left( \sum_{s=1}^{S} \sum_{i=1}^{f_l} U \text{diag} \Big( W_{i,j}^{(l,s)} \Phi_s(\boldsymbol{\lambda}) \Big) U^\top H_i^{(l)} \right). \tag{15}$$

Then, we take out the scalars $W_{i,j}^{(l,s)}$ of the diag operator:

$$H_j^{(l+1)} = \sigma \left( \sum_{s=1}^{S} \sum_{i=1}^{f_l} W_{i,j}^{(l,s)} U \text{diag}(\Phi_s(\boldsymbol{\lambda})) U^\top H_i^{(l)} \right). \tag{16}$$

Let us define a convolution operator $C^{(s)} \in \mathbb{R}^{n \times n}$ as:

$$C^{(s)} = U \text{diag}(\Phi_s(\boldsymbol{\lambda})) U^\top. \tag{17}$$

Using equation 16 and equation 17, we have thus:

$$H_j^{(l+1)} = \sigma \left( \sum_{i=1}^{f_l} \sum_{s=1}^{S} W_{i,j}^{(l,s)} C^{(s)} H_i^{(l)} \right). \tag{18}$$

Then, each term of the sum over $s$ corresponds to a matrix $H^{(l+1)} \in \mathbb{R}^{n \times f_{l+1}}$ with

$$H^{(l+1)} = \sigma \left( C^{(1)} H^{(l)} W^{(l,1)} + \cdots + C^{(S)} H^{(l)} W^{(l,S)} \right), \tag{19}$$

with $H^{(l)} = [H_1^{(l)}, \ldots, H_{f_l}^{(l)}]$. We get by grouping the terms:

$$H^{(l+1)} = \sigma \left( \sum_{s=1}^{S} C^{(s)} H^{(l)} W^{(l,s)} \right), \tag{20}$$

which corresponds to equation 3. Therefore, equation 5 corresponds to equation 3 with $C^{(s)}$ defined as equation 6. $\qquad \square$

## C   PROOF OF COROLLARY 1.1

*Proof.* By using equation 6 from Theorem 1, we can obtain a spatial convolution kernel $C^{(s)}$ whose frequency profile is $\Phi_s(\boldsymbol{\lambda})$. Since the eigenvector matrix is orthonormal (i.e., $U^{-1} = U^\top$), we can extract $\Phi_s(\boldsymbol{\lambda})$, which yields equation 7. $\qquad \square$

## D   PROOF OF THEOREM 2

ChebNet relies on the approximation of a spectral graph analysis proposed in (Hammond et al., 2011), based on the Chebyshev polynomial expansion of the scaled graph Laplacian. Even though its multi supports frequency responses are known, to the sake of simplicity, it was represented in form of equation 3 in (Defferrard et al., 2016) as follows;

$$C^{(1)} = I, \;\; C^{(2)} = 2L/\lambda_{\max} - I, \;\; C^{(k)} = 2C^{(2)} C^{(k-1)} - C^{(k-2)}. \tag{21}$$

*Proof.* When the identity matrix is used as convolution kernel, it just directly transmits the inputs to the outputs without any modification. This process is called all-pass filter. Mathematically, we can calculate the full frequency profile for kernel $I$ by using Corollary 1.1, namely

$$\Phi_1 = U^\top I U = U^\top U = I, \tag{22}$$

since the eigenvectors are orthonormal. Therefore, we can parametrize the diagonal of the full frequency profile by $\boldsymbol{\lambda}$ and reach the standard frequency profile for the first ChebNet support as follows:

$$\Phi_1(\boldsymbol{\lambda}) = \text{diag}(I) = \mathbf{1}. \tag{23}$$

We can compute the $C^{(2)}$ kernel full frequency profile using Corollary 1.1:

$$\Phi_2 = U^\top \left( \frac{2}{\lambda_{\max}} L - I \right) U. \tag{24}$$

Since $U^\top I U = I$, equation 24 can be rearranged as

$$\Phi_2 = \frac{2}{\lambda_{\max}} U^\top L U - I. \tag{25}$$

Since $\boldsymbol{\lambda} = [\lambda_1, \ldots, \lambda_n]$ are the eigenvalues of the graph Laplacian $L$, those must conform to the following condition:

$$LU = U\text{diag}(\boldsymbol{\lambda}); \tag{26}$$
$$U^\top LU = \text{diag}(\boldsymbol{\lambda}). \tag{27}$$

Replacing equation 27 into equation 25, we get

$$\Phi_2 = \frac{2}{\lambda_{\max}}\text{diag}(\boldsymbol{\lambda}) - I. \tag{28}$$

This full frequency profile consists of two parts, a diagonal matrix and the negative identity matrix. Therefore, we can parametrize the full frequency matrix diagonal to show the standard frequency profile as follows:

$$\Phi_2(\boldsymbol{\lambda}) = \text{diag}(\Phi_2) = \frac{2\boldsymbol{\lambda}}{\lambda_{\max}} - \mathbf{1}. \tag{29}$$

Given the third and following ChebNet supports, when we use Corollary 1.1, the corresponding frequency profile is

$$\Phi_k = U^\top \left(2C^{(2)}C^{(k-1)} - C^{(k-2)}\right) U. \tag{30}$$

By expanding equation 30, we get

$$\Phi_k = 2U^\top C^{(2)}C^{(k-1)}U - U^\top C^{(k-2)}U. \tag{31}$$

Since $UU^\top = I$, we can insert the product $UU^\top$ into equation 31. Thus, we have

$$\Phi_k = 2U^\top C^{(2)}UU^\top C^{(k-1)}U - U^\top C^{(k-2)}U \tag{32}$$
$$\Phi_k = 2\left(U^\top C^{(2)}U\right)\left(U^\top C^{(k-1)}U\right) - U^\top C^{(k-2)}U. \tag{33}$$

Since $\Phi_{k'} = U^\top C^{(k')}U$ for any $k'$, it yields:

$$\Phi_k = 2\Phi_2\Phi_{k-1} - \Phi_{k-2}, \tag{34}$$

Hence $\Phi_1$ and $\Phi_2$ are diagonal matrices, and the rest of the kernels frequency profiles become diagonal matrices in equation 34. Therefore, we can write the corresponding standard frequency profiles of third and following ChebNet convolution supports as follows:

$$\Phi_k(\boldsymbol{\lambda}) = 2\Phi_2(\boldsymbol{\lambda})\Phi_{k-1}(\boldsymbol{\lambda}) - \Phi_{k-2}(\boldsymbol{\lambda}). \tag{35}$$

$\square$

# E    PROOF OF THEOREM 3

*Proof.* CayleyNet was originally defined as it uses the weight vector parametrization of $F_i^{(l,j)} = [g_{i,j,l}(\lambda_1, h), \ldots, g_{i,j,l}(\lambda_n, h)]^\top$ in equation 1, where the function $g(\cdot, \cdot)$ is defined in (Levie et al., 2019) by

$$g(\lambda, h) = c_0 + 2Re\left(\sum_{k=1}^{r} c_k \left(\frac{h\lambda - \mathbf{i}}{h\lambda + \mathbf{i}}\right)^k\right), \tag{36}$$

where $\mathbf{i}^2 = -1$, $Re(\cdot)$ is the function that returns the real part of a given complex number, $c_0$ is a trainable real coefficient, and $c_1, \ldots, c_r$ are complex trainable coefficients. We can write $h\lambda - \mathbf{i}$ in Euler form by $\sqrt{h^2\lambda^2 + 1}.e^{\mathbf{i}atan2(-1,h\lambda)}$ and for $h\lambda + \mathbf{i}$ by $\sqrt{h^2\lambda^2 + 1}.e^{\mathbf{i}atan2(1,h\lambda)}$. By this substitution, equation 36 becomes

$$g(\lambda, h) = c_0 + 2Re\left(\sum_{k=1}^{r} c_k e^{\mathbf{i}k(atan2(-1,h\lambda) - atan2(1,h\lambda))}\right). \tag{37}$$

where $atan2(y, x)$ is the inverse tangent function, which finds the angle (in range of $[-\pi, \pi]$) of a point given its $y$ and $x$ coordinates. For further simplification, let us introduce the $\theta(\cdot)$ function defined by

$$\theta(x) = atan2(-1, x) - atan2(1, x). \tag{38}$$

Since the $c_k$s are complex numbers, we can write them as a sum of real and imaginary parts, $c_k = a_k/2 + \mathbf{i}b_k/2$ (the scale factor 2 is added for convenience). Thus, equation 37 can be rewritten as follows:

$$g(\lambda, h) = c_0 + Re\left(\sum_{k=1}^{r}(a_k + \mathbf{i}b_k)e^{\mathbf{i}k\theta(h\lambda)}\right). \tag{39}$$

We can replace $e^{\mathbf{i}k\theta(h\lambda)}$ with its polar coordinate equivalence form $\cos(k\theta(h\lambda)) + \mathbf{i}\sin(k\theta(h\lambda))$. When we remove the imaginary components because of $Re(\cdot)$ function, equation 39 becomes

$$g(\lambda, h) = c_0 + \sum_{k=1}^{r} a_k \cos(k\theta(h\lambda)) - b_k \sin(k\theta(h\lambda)). \tag{40}$$

In this definition, there is no complex coefficient, but only real coefficients ($c_0$, $a_k$ and $b_k$ for $k = 1, \ldots, r$) to be tuned by training. By using the form in equation 40, we can parametrize CayleyNet by the parametrization matrix $B \in \mathbb{R}^{n \times 2r+1}$ by

$$[g(\lambda_0, h), \ldots, g(\lambda_n, h)]^\top = B[c_0, a_1, b_1, \ldots, a_r, b_r]^\top. \tag{41}$$

The $s$-th column vector of matrix $B$, denotes $B_s$, must fulfill the following conditions:

$$B_s = \Phi_s(\boldsymbol{\lambda}) = \begin{cases} \mathbf{1} & \text{if } s = 1 \\ \cos(\frac{s}{2}\theta(h\boldsymbol{\lambda})) & \text{if } s \in \{2, 4, \ldots, 2r\} \\ -\sin(\frac{s-1}{2}\theta(h\boldsymbol{\lambda})) & \text{if } s \in \{3, 5, \ldots, 2r+1\} \end{cases} \tag{42}$$

$\square$

We can see CayleyNet as a spectral graph convolution that uses $2r + 1$ convolution kernels. The first kernel is an all-pass filter, and the frequency profiles of remaining $2r$ kernels ($\Phi_s(\boldsymbol{\lambda})$) are created using sine and cosine functions, with a parameter $h$ used to scale the eigenvalues in equation 42. Considering equation 6 in Theorem 1, we can write CayleyNet's convolutions ($C^{(s)}$) in spatial domain. CayleyNet includes the tuning of this scaling parameter in the training pipeline. Note that because of the function definition in equation 38, $\theta(h\lambda)$ is not linear in $\lambda$. Therefore, $\Phi_s$ cannot be a perfect sinusoidal in $\lambda$s.

## F  PROOF OF THEOREM 4

One major simplification of the ChebNet is Graph Convolution Network (GCN) (Kipf & Welling, 2017), which has single convolution support and already presented in our framework in equation 3. The first proposal of this paper uses the subtraction of the second ChebNet support from the first one under the assumption of $\lambda_{\max} = 2$ and $L$ is the normalized graph Laplacian, as it is defined by $C_{GCN^*} = C^{(1)} - C^{(2)} = 2I - L$.

**Proposition 1.** $C_{GCN^*} = 2I - L$ is a spectral-designed support and its frequency response is $\Phi_{GCN^*}(\boldsymbol{\lambda}) = 2 - \boldsymbol{\lambda}$.

*Proof.* If the assumption is true, it should meet:

$$2I - L = U\text{diag}(2 - \boldsymbol{\lambda})U^\top \tag{43}$$

this can be written in the following way as well

$$2I - L = 2UIU^\top - U\text{diag}(\boldsymbol{\lambda})U^\top \tag{44}$$

since $UIU^\top = I$ and $U\text{diag}(\boldsymbol{\lambda})U^\top = L$, the proposition is true. $\square$

One can see that the first proposal of GCN is spectral-designed and it is low-pass filter. That is why GCN is misclassified as a spectral approach in the literature (Wu et al., 2019b; Chami et al., 2020). However, instead of using this version, GCN used re-normalization trick and defined its final single convolution support as:

$$C_{GCN} = (D + I)^{-1/2}(A + I)(D + I)^{-1/2}, \tag{45}$$

where $D$ is diagonal degree matrix and $A$ is the adjacency matrix.

**Proposition 2.** $C_{GCN} = (D + I)^{-1/2}(A + I)(D + I)^{-1/2}$ *frequency response is* $\Phi_{GCN}(\boldsymbol{\lambda}) = \mathbf{1} - \frac{p}{p+1}\boldsymbol{\lambda}$ *for regular graphs whose node degrees are p.*

*Proof.* When all node degrees are $p$, we can write diagonal degree matrix by $D = pI$. It yields, $L = I - A/p$ or $A = pI - pL$. When we substitute new equations of $A$ and $D$ into GCN support, we get

$$C_{GCN} = \frac{pI - pL + I}{p + 1} = I - \frac{p}{p+1}L. \tag{46}$$

It should meet the following condition if the given frequency response is true:

$$I - \frac{p}{p+1}L = U\mathrm{diag}(\mathbf{1} - \frac{p}{p+1}\boldsymbol{\lambda})U^\top \tag{47}$$

Since $U\mathrm{diag}(\mathbf{1})U^\top = I$, and $U\mathrm{diag}(\boldsymbol{\lambda})U^\top = L$, the condition in equation 47 is satisfied. □

This proposition shows that the GCN frequency profile acts as a low-pass filter. When the given graph is a regular graph, all node degrees are equal for the case of $p = 2$, is leading to a frequency profile defined by $\mathbf{1}-2\boldsymbol{\lambda}/3$. Since the normalized graph Laplacian eigenvalues are in the range $[0, 2]$, the filter magnitude linearly decreases until the third quarter of the spectrum (cut-off frequency) where it reaches zero. Then it linearly increases until the end of the spectrum. This explains the shape of the frequency profile of GCN convolutions for 1D regular graph observed in Figure 2a in blue one.

However, this conclusion cannot explain the perturbations on the GCN frequency profile. To analyse this point, we relax the assumption $D = pI$ and rewrite equation 45 as follows and start to proof.

$$C_{GCN} = (D + I)^{-1} + (D + I)^{-1/2}A(D + I)^{-1/2}. \tag{48}$$

*Proof.* We can see that the GCN kernel consists of two parts, $C_{GCN} = c_1 + c_2$, where first part is given by $c_1 = (D + I)^{-1}$ and the second one is $c_2 = (D + I)^{-1/2}A(D + I)^{-1/2}$.

For the second part ($c_2$), we can write it using the element-wise multiplication operator $\odot$ (Hadamard multiplication)

$$c_2 = A \odot \sqrt{\mathbf{1}/(d + 1)} \cdot \sqrt{\mathbf{1}/(d + 1)}^\top, \tag{49}$$

where $d$ is the column degree vector $d = \mathrm{diag}(D)$ and the division and square root are also element-wise (Hadamard) operations. With the same notation, we can rewrite the Chebyshev second kernel, assuming that $\lambda_{\max} = 2$,

$$C^{(2)} = -A \odot \sqrt{\mathbf{1}/d} \cdot \sqrt{\mathbf{1}/d}^\top. \tag{50}$$

The two expressions equation 49 and equation 50 show that negative $c_2$ is an approximation of the second Chebyshev kernel if vector $d$ consists of same values, as it was assumed in Proposition 2. When the vector $d$ is composed of different values, the two matrices $\sqrt{\mathbf{1}/d}.\sqrt{\mathbf{1}/d}^\top$ and $\sqrt{\mathbf{1}/(d + 1)}.\sqrt{\mathbf{1}/(d + 1)}^\top$ are not proportional for each coordinate (i.e., entry). To obtain $c_2$ from $C^{(2)}$, we need to use different coefficients for each coordinate of the kernel. If the difference between node degrees is important, these coefficients have the strong influence, and $c_2$ may be very different from $C^{(2)}$. Conversely, if the node degrees are quite uniform, these coefficients may be neglected. This phenomenon is the first cause of perturbation on GCN frequency profile.

The first part ($c_1$) of the GCN kernel in equation 48 is more interesting. Actually, it is a diagonal matrix that shows the contribution of each node in the convolution process. Instead of looking for some approximations of known frequency profiles such as those of Chebyshev kernels, we can write

its frequency profile directly. Using Corollary 1.1, we can express the frequency profile of $c_1$ in matrix form by

$$\Phi_{c_1} = (U^\top c_1 U), \tag{51}$$

where $U$ is the eigenvectors matrix. By taking advantage of having a diagonal kernel $c_1$, we can express each component of full frequency profile as

$$(\Phi_{c_1})_{i,j} = \sum_{k=1}^{n} \left( \frac{1}{1 + d_k} U_{i,k} U_{j,k} \right), \tag{52}$$

where $n$ is the number of nodes in the graph, $d_k$ is degree of the $k$-th node, $U_{i,k}$ is the $k$-th element of $i$-th eigenvector. As eigenvectors $U_i$ and $U_j$ are orthogonal for $i \neq j$, their scalar product is null. However, in equation 52, the weighting coefficient $\frac{1}{1+d_k}$ is not constant over all the dimensions of the eigenvectors. Therefore, there is no guarantee that $\Phi_{c_1}(i, j)$ is null. This is another reason that explains that the GCN frequency profile has many non-zero elements outside of the diagonal.

In addition, it is also clear that the standard frequency profile of $c_1$ (diagonal of $\Phi_{c_1}$, i.e., $(\Phi_{c_1})_{i,i}$ in equation 52) is not smooth. Indeed, the diagonal elements of $\Phi_{c_1}$ can be written as a weighted sum of squared eigenvalues elements, which again is weighted by $1/(1 + d_k)$. If the latter is constant for all $k$, the sum of squared eigenvectors elements has to be 1 since the eigenvectors have unit L2-norm. But in the general case where $1/(1 + d_k)$ are not necessarily constant over all the dimensions of eigenvectors, the diagonal of the matrix may have some perturbations. This point constitutes another explanation on the fact that the GCN standard frequency profile is not smooth.

On the other hand, under the assumption that the node degrees distribution is uniform, we can derive the following approximation:

$$p \approx \overline{p} = \frac{1}{n} \sum_{k=1}^{n} d_k. \tag{53}$$

We can then write an approximation of the GCN frequency profile as a function of the average node degree by replacing $p$ with $\overline{p}$ and obtain the final approximation:

$$\Phi_{GCN}(\boldsymbol{\lambda}) \approx \mathbf{1} - \frac{\overline{p}}{\overline{p} + 1} \boldsymbol{\lambda}. \tag{54}$$

$\square$

Therefore, we can theoretically show the cut-off frequency, namely where GCN kernel frequency profile reaches 0, by

$$\lambda_{\text{cut}} \approx \frac{\overline{p} + 1}{\overline{p}}. \tag{55}$$

## G  PROOF OF THEOREM 5

Graph Isomorphism Network (GIN) defined in (Xu et al., 2019) has a single convolution support defined as follows:

$$C_{GIN} = A + (1 + \epsilon)I, \tag{56}$$

where $\epsilon$ is a trainable parameter that makes the support trainable (GIN-$\epsilon$) and classified as spatial-designed trainable-support graph convolution. Another version named GIN-0 is also defined in the same paper where $\epsilon = 0$, which makes $C_{GIN} = A + I$; thus, the convolution becomes fixed-support and identical with Vanilla GNN defined in Section 2.2.

The proof of Theorem 5 relies on the following proposition.

**Proposition 3.** *For $C_{GIN} = A + (1+\epsilon)I$, the frequency response is $\Phi_{GIN}(\boldsymbol{\lambda}) = p \left( \frac{1+\epsilon}{p} + 1 - \boldsymbol{\lambda} \right)$ for regular graphs, where $p$ is the node degrees.*

*Proof.* When all node degrees are $p$, it yields $D = pI$, $L = I - A/p$ or $A = pI - pL$. When we substitute the expressions of $A$ and $D$ into $C_{GIN}$, we get

$$C_{GIN} = (p + 1 + \epsilon)I - pL. \tag{57}$$

It should meet the following condition if the given frequency response is true:

$$(p + 1 + \epsilon)I - pL = U\text{diag}(p + \epsilon + \mathbf{1} - p\boldsymbol{\lambda})U^\top. \tag{58}$$

We can obtain the following equation by $p + \epsilon + \mathbf{1} = (p + 1 + \epsilon)I$ substitution:

$$(p + 1 + \epsilon)I - pL = (p + 1 + \epsilon)UIU^\top - pU\text{diag}(\boldsymbol{\lambda})U^\top. \tag{59}$$

Since $UIU^\top = I$ and $U\text{diag}(\boldsymbol{\lambda})U^\top = L$, the condition in equation 58 is satisfied. □

By relying on Proposition 58, we establish Theorem 5 as follows.

*Proof.* Even in regular graph, the theoretical frequency response of GIN is not identical and it depends on the node degree, thus it is not spectral-designed. In addition, we can see the GIN convolution support as the sum of two matrices where the second one $(1 + \epsilon)I$ is diagonalizable by eigenvectors $U$ of graph Laplacian by $\Phi = \mathbf{1} + \epsilon$. Thus, the second part of GIN support is spectral. However, the first part, which is adjacency $A$, cannot be diagonalizable by $U$. Since the convolution support is not diagonalizable, we cannot write exact frequency response of GIN convolution but just an approximation of Proposition 3, assuming by the average node degree of the graph is $\overline{p}$ in

$$\Phi_{GIN}(\boldsymbol{\lambda}) \approx \overline{p}\left(\frac{1 + \epsilon}{\overline{p}} + \mathbf{1} - \boldsymbol{\lambda}\right). \tag{60}$$

□

## H  ADDITIONAL RESULTS ON SPECTRAL ANALYSIS

### H.1  CHEBNET

To get empirical frequency responses of ChebNet supports, we used regular 1D graph, Cora, Cite-Seer. As confirmed in theoretical analysis, in all cases the frequency responses do not depend on graph. The magnitude (in absolute value) of the frequency responses are shown in Figure 1a. As stated by Theorem 2, the first two kernel frequency profiles of ChebNet are $\Phi_1(\boldsymbol{\lambda}) = \mathbf{1}$ and $\Phi_2(\boldsymbol{\lambda}) = 2\boldsymbol{\lambda}/\lambda_{\max} - \mathbf{1}$, where $\mathbf{1}$ is the vector of ones. Since $\lambda_{\max} = 2$ for all graphs that we used in the analysis, we get $\Phi_2(\boldsymbol{\lambda}) = \boldsymbol{\lambda} - \mathbf{1}$. The third one and following kernel frequency profiles can also be computed using $\Phi_k(\boldsymbol{\lambda}) = 2\Phi_2(\boldsymbol{\lambda})\Phi_{k-1}(\boldsymbol{\lambda}) - \Phi_{k-2}(\boldsymbol{\lambda})$, leading to $\Phi_3(\boldsymbol{\lambda}) = 2\boldsymbol{\lambda}^2 - 4\boldsymbol{\lambda} + \mathbf{1}$ for example for the third kernel. One can easily confirm the functions in range of $[0...2]$ by relevant plot in Figure 1a.

Thanks to Chebyshev polynomial expansion, we do not need to calculate supports by eigendecomposition which makes the method computationally efficient. Besides, as it is spectral designed, ChebNet covers all the spectrum. Theoretically it can create all necessary filters if we use many kernels and stack the layers back to back. However higher order supports frequency responses are less smooth than lower order ones. This does not guarantee that the graph convolution transferability is maintained. For this reason, in literature, generally a few (up to first 3) supports are used (Defferrard et al., 2016; Kipf & Welling, 2017).

### H.2  CAYLEYNET

Since CayleyNet is spectral, its supports frequency response is consistent and does not change according to graph structure. It leads to get the same empirical results for all our attempts on 1D graph, Cora, CiteSeer graphs. Theorem 3 result can be compared to relevant support result in Figure 1b. For instance, the first support frequency response is $\Phi_1(\lambda) = 1$ as it is all-pass filter in Figure 1b (blue plot). If we assume zoom parameter is $h = 1$, the second support frequency response becomes $\Phi_2(\lambda) = cos(\theta(\lambda))$. To confirm that result, we can first check the case where $\lambda = 0$. Since $\theta(0) = -\pi$, $\Phi_2(0) = -1$ where its magnitude (absolute value) in the diagram is 1. Later, we can check the case where $\lambda = 1$. Since $\theta(1) = -\pi/2$, thus $\Phi_2(1) = 0$ as seen in the orange plot in Figure 1b, $\lambda = 1$ is the cut-off frequency for the second support of CayleyNet.

Having multi-support with different frequency responses makes the convolution productive in terms of output signal profile. Moreover, by learning zoom parameter, theoretically, we can shrink (higher

$h$ value) or expand (smaller $h$ value) the frequency responses which is needed according to the problem. However, it makes the supports non-static. The supports need to be calculated in each learning epoch. Although, to limit the induced computational cost, an approximation is computed using a fixed number of Jacobi iterations (Levie et al., 2019). But still, it seems not efficient in benchmark problem. Instead, a fixed value for the $h$ parameter might be used and $h$ can be treated as an hyperparameter to be tuned according to validation set. Besides, it has no band specific supports, but band-pass might be obtained by using multi supports and stacked layers.

## H.3 GCN

By a clear margin, the most popular method is GCN (Kipf & Welling, 2017) in GNN literature, thanks to its simplicity and relatively good results on some benchmark dataset. However, as we prove in Theorem F, it is a low-pass filter. Since it has a single support, one kind of filter which is low-pass, stacking that layer in deep architecture will not work. Because it continuously smooths the signal on the graph and on the final layer, there will only be a smoothed signal.

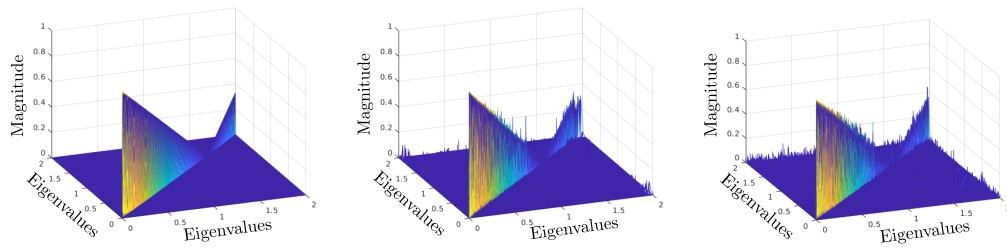

Figure 5: Full frequency response of GCN on 1D, Cora and CiteSeer graphs

The three standard frequency responses in Figure 2a have almost the same low-pass filter shape. It corresponds to a function composed of a decreasing part on the three first quarters of the eigenvalues range, followed by an increasing part on the remaining range. This observation is coherent with the theoretical analysis. Hence, kernels used in GCN are transferable across the three graphs at hand. In Figure 2a, the cut-off frequency of the 1-D linear circular graph is exactly 1.5, while it is about 1.35 for CiteSeer. This observation can be explained by the fact that when considering a 1-D linear circular graph, all nodes have a degree ($\bar{p} = 2$), hence $\lambda_{\text{cut}} = 1.5$. Since the average node degree in CiteSeer is 2.77, therefore $\lambda_{\text{cut}} \approx 1.36$.

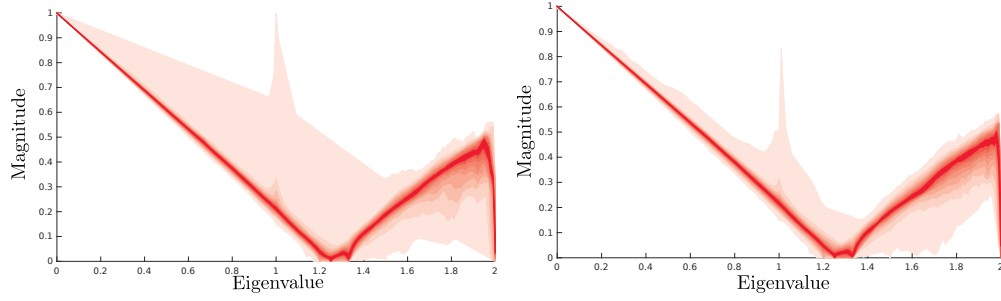

Figure 6: Heat map of GCN's frequency profiles on ENZYMES and PROTEIN dataset graphs.

Concerning the full frequency responses, there is no contribution outside the diagonal for the regular line graph (Figure 5 a). Conversely, some off-diagonal values are not null for CiteSeer and Cora (Figure 5 b-c). Again, this observation confirms the theoretical analysis in Appendix F. We also provided heat map of frequency responses of GCN convolution on more realistic biological graph dataset named ENZYMES and PROTEIN in Figure 6. The majority of the graph frequency

responses have almost the same shape. However, some graph frequency responses are far away from the expected frequency response as illustrated with lighter color in heatmap.

## H.4  GIN

GIN model defined in Xu et al. (2019) is not just a layer but a mini multi-layer model. The first layer is the main graph convolution layer which we analyzed, followed by at least one but preferably two MLP layers. According to Theorem 5, the frequency response of the main GIN convolution has $1 - \epsilon + \overline{p}$ of magnitude at zero eigenvalue and it decreases with respect to the eigenvalue. The outer node degree is a scaling factor of the frequency response that does not have any effect of the character of filter. While $\epsilon$ increases, the cut-off frequency of the convolution increases, thus it makes more low-pass effect. On the other hand, when $\epsilon$ decreases, the cut-off decreases, thus it makes more high-pass effect while inner $\overline{p}$ can be seen as multiplicand of the effect. Theoretically, we can say that GIN's cut-off frequency is $\lambda_{\text{cut}} \approx 1 + (1 + \epsilon)/\overline{p}$, which is the same with GCN, if $\epsilon = 0$.

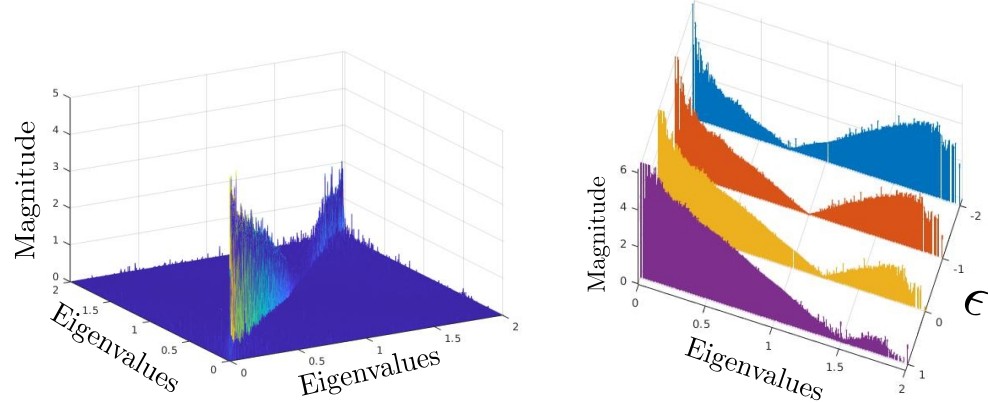

Figure 7: Full frequency profiles of GIN-0 and frequency responses of different $\epsilon$ values for Cora graph.

One can easily calculate the frequency response of adjacency as a convolution support where $C = A$. It can be seen as a special GIN support where $\epsilon = -1$. Thus $\Phi_A(\boldsymbol{\lambda}) \approx \overline{p}(\mathbf{1} - \boldsymbol{\lambda})$. The formulation is almost the same as the one given by (NT & Maehara, 2019). It differs by the scaling factor and an approximation of the regular graph case. Since the eigenvalues of the normalized Laplacian lie on the interval [0, 2], it works as a notch-like band-stop filter for intermediate frequency ($\lambda = 1$). But, in most of applications, the eigenvalues greater than 1 is less likely. In this case, there are less component to pass. It results that using adjacency has more likely a low-pass effect as concluded in (NT & Maehara, 2019).

The experimental analysis of the spectral behavior of GIN (Xu et al., 2019) first implies to compute the convolution kernel as given in equation 56 for $\epsilon = \{-2, -1, 0, 1\}$. Then, the spectral representation of the obtained convolution matrix can be calculated using Corollary 1.1. This result leads to the frequency profiles illustrated in Figure 2b-c (1D and CiteSeer graph) and Figure 7 (Cora graph). For regular 1D circular graph where $p = 2$, the frequency responses are absolutely what Proposition 3 indicates. As shown in Figure 2b, the cut-off frequencies are 2, 1.5, 1.0, and 0.5 for $\epsilon = \{1, 0, -1, -2\}$ respectively. But for realistic graphs such as CiteSeer, since its average degree is $\overline{p} \approx 2.77$, the cut-off frequencies are 1,72, 1.36, 1.0 and 0.63 for $\epsilon = \{1, 0, -1, -2\}$ respectively as shown in Figure 2c. The results for Cora graph are slightly different than CiteSeer in Figure 7b because of the fact that average node degree is different. Since GIN does not have spectral designed support, in its full frequency profile, there are some non-zero components out of the diagonal as shown in Figure 7a for GIN-0 model on Cora. Figure 8 and 9 show the heat map of frequency responses of the GIN model under $\epsilon = \{1, 0, -1, -2\}$ on ENZYMES and PROTEIN collection of graphs.

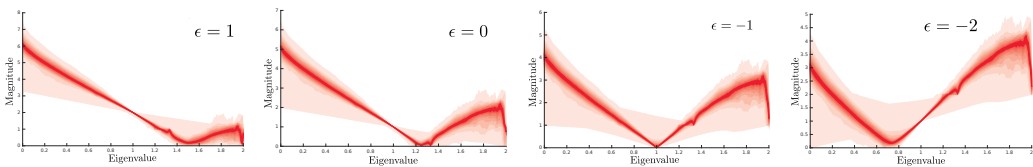

Figure 8: Heat map of different $\epsilon$ valued GIN frequency profiles on ENZYMES

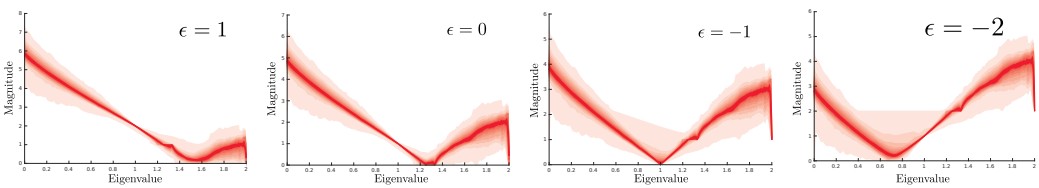

Figure 9: Heat map of different $\epsilon$ valued GIN frequency profiles on PROTEIN

### H.5 GAT

One can see that, each support is a function of trainable weights $(W^{(l,s)}, \mathbf{a}^{(l,s)})$ in GAT, frequency profiles cannot be directly computed similarly to previous ones. We did bunch of simulations for Cora graph. As the proposed method used for Cora problem, we have generated 8 different convolution supports corresponding to 8 pairs of $W^{(l,s)} \in R^{1433 \times 8}$ (1433 features for each node) and $\mathbf{a}^{(l,s)} \in R^{16 \times 1}$ trainable weights for the first layer (Veličković et al., 2018). We produce 240 (30 for each support) random pairs of $W^{(l,s)}$ and $\mathbf{a}^{(l,s)}$ where activation function is `LeakyReLU` has 0.2 negative slope as in (Veličković et al., 2018). Later, we calculated frequency response of generated supports by Corollary 1.1. The mean and standard deviation of the frequency profiles for these simulated GAT supports are shown in Figure 3 a and its expected and standard deviation of the full frequency response shown in Figure 10.

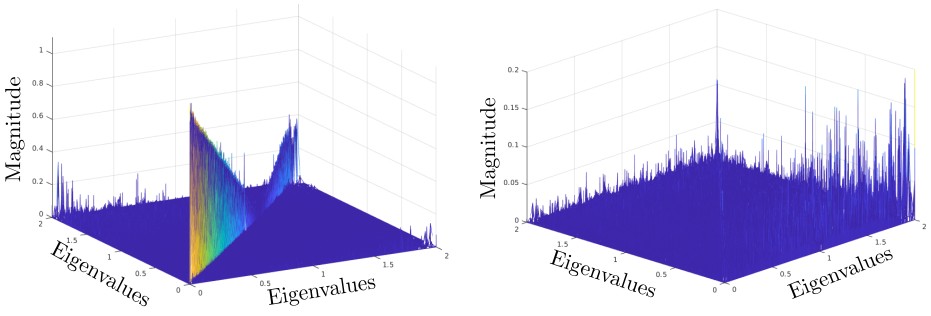

Figure 10: Full frequency profile of GAT and its standard deviation.

The full frequency profile is not symmetric as seen in Figure 10a. According to Figure 10b, variations are mostly on the right side of the diagonal in the full frequency profile. This is related to the fact that these convolution kernels are not symmetric. However, the variation on frequency profile might not be sufficient in problems that need some specific band-pass filters.

In order to get trained attention head frequency responses in ENZYMES and PROTEIN datasets, we randomly divided the dataset into 4 folds. We trained the 2-layer GAT model which has 25 attention heads each by using 3 folds. We calculated attention heads (50 $C$ matrices given in equation 12, 25 each layer) for each graph in test fold. The density heat map in Figure 3b-c are the frequency responses of these attention heads for ENZYMES and PROTEINS dataset respectively.

Table 3: Sum of squared errors. All models have roughly 30k trainable parameters.

| Prediction Target | GCN | GIN | GAT | ChebNet |
|---|---|---|---|---|
| Low-pass filter ($\Phi_1$) | 15.55 | 11.01 | 10.50 | 3.44 |
| Band-pass filter ($\Phi_2$) | 79.72 | 63.24 | 79.68 | 17.30 |
| High-pass filter ($\Phi_3$) | 29.51 | 14.27 | 29.10 | 2.04 |

# I  WHICH FILTERS CAN THE GNN MODELS LEARN?

In this section, we seek to measure of the ability of GNN models to learn some specific filtering process. This study is very important in order to understand the learning capability of existing GNN models. Since the problem may need various types of filtering, the best GNN model has to be able to learn any kind of filtering.

For this purpose, we conduct an empirical analysis on a real image with resolution of $100{\times}100$ and its corresponding 2D regular 4-neighborhood grid graph. The input of the GNN is the adjacency matrix of size $10000{\times}10000$ and the pixel intensities given in a 10000-length vector. We create three different spectral filters that correspond to low-pass, band-pass and high-pass effects and apply these filters to the given input image. Our selection of spectral filters are defined by $\Phi_1(\rho) = \exp(-100\rho^2)$, $\Phi_2(\rho) = \exp(-1000(\rho - 0.5)^2)$ and $\Phi_3(\rho) = 1 - \exp(-10\rho^2)$ for low-pass, band-pass and high-pass filters respectively, where $\rho^2 = u^2 + v^2$ and $u$ and $v$ are the normalized frequencies on each direction for a given image resolution. Used input image and its filtering results can be found in Figure 11.

Since we do not use pixel positions, neither as node feature nor as edge feature, we create these spectral filters to be learned in a directional agnostic way. Therefore, the problem can be viewed as a single graph node regression problem, where we train the GNN models to minimize the square error between its output and targeted filtered image.

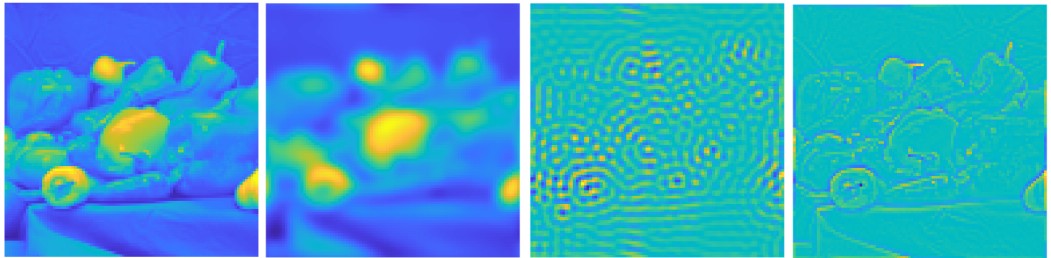

Figure 11: Input image, and its filtering results by $\Phi_1$, $\Phi_2$ and $\Phi_3$ respectively

In order to assess ChebNet, GCN, GIN and GAT, we use a 3-layer GNN architecture whose input is a one-length feature (intensity of the pixel) and the number of neurons in hidden layers is respectively 32, 64 and 64; the output layer is an MLP that projects the final node representation onto the single output for each node. We used roughly 30k trainable parameter in ChebNet with 5 supports. For the other methods, we tuned the hidden neuron numbers in order to be sure that they have a similar number of trainable parameters. Since the aim is not assessing the generalization performance, we do not use any regularization or dropout to address overfitting, but simply force the GNN to learn the input-output relation. We keep the iterations till there is no improvement for consecutive 100 iterations or maximum 3000 iterations.

Table 3 gives the sum of squared errors between target and the output of the trained model. One can see that ChebNet constantly outperformed GCN, GIN and GAT for all tasks. For learning low-pass filtering, the rest of the models did better compared to the high-pass and band-pass tasks. That is the fact that GCN, GIN and GAT have the ability to act as low-pass filters. In addition to do better on the low-pass task, GIN also did relatively better on the high-pass task as well. It is obvious that GIN can work as high pass if the $\epsilon$ parameter is selected negative (see Theorem 5). It turns out that the trained values of $\epsilon$ in GIN for each layer are $-5.27$, $-2.21$ and $-0.47$ for the high-pass task.

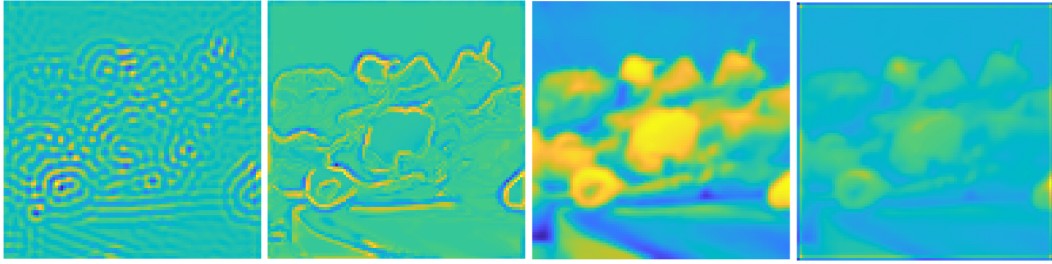

Figure 12: The output of GNNs trained with band-pass task. Images are taken from ChebNet, GIN, GAT and GCN respectively.

Table 4: ChebNet's sum of squared errors on band-pass tasks with respect to $S$ kernels and $L$ stacked layers. All models have roughly 30k trainable parameters.

|       | $S=2$ | $S=3$ | $S=5$ | $S=7$ | $S=10$ |
|-------|-------|-------|-------|-------|--------|
| $L=1$ | 65.06 | 56.25 | 47.12 | 39.59 | 28.15  |
| $L=2$ | 55.85 | 45.96 | 26.81 | 18.92 | 13.33  |
| $L=3$ | 50.13 | 36.60 | 17.30 | 9.84  | 7.32   |
| $L=4$ | 44.88 | 24.89 | 11.90 | 8.91  | 6.96   |

Thanks to the spectral-designed convolution supports in ChebNet, it could learn high-pass and low-pass tasks very well. However for band-pass tasks, even though it is the best in this category too, it still has large errors compared to the high-pass and low-pass tasks. This is due to the fact that the selected band-pass filter is very narrow, because the coefficient $-1000$ in the formulation of $\Phi_2$ makes the used ChebNet (with 5 convolution supports and 3 layers) unable to adapt this stiff (not smooth) filter function. Moreover, since ChebNet has no band specific convolutions, band specific output can be produced if the number of kernels increases (going wider) and/or the model goes deeper. To clarify this point, we conducted another test for band-pass task on ChebNet to show the effect of going deeper in the model and going wider (increase the convolution support) while keeping the trainable parameters fixed. These results are given in Table 4.

According to Table 4, the ability of ChebNet to learn the given frequency response becomes better with respect to the number of convolution supports and number of layers. However, this result is not surprising where it is proved that any frequency response can be written by a weighed sum of enough number of Chebyshev polynomials (Hammond et al. (2011)). When we train the ChebNet, it just finds these coefficients to create the target frequency response by minimizing the error. However, the interesting point is the incapability of GCN, GIN and GAT methods to even create reasonable approximations of these targeted filter effects. For instance, it can be seen in Figure 12 that ChebNet performed well to produce the desired band-pass output. However, GAT and GCN produce just a different kind of low-pass filtering result instead of band-pass, while GIN at least can find edges (high-pass component) thank to its trainable parameter $\epsilon$. We also tested the deeper network for GCN, GAT and GIN as well and have not seen any significant improvement when we use deeper network.

## J    CAN GNN CLASSIFY GRAPHS ACCORDING TO FREQUENCY OF ITS SIGNAL?

In this section, we measure the generalization ability of GNN for graph classification problem where graph classes depend on the signal that the graphs carry. We generate 5000 images of $100{\times}100$ pixels composed of random generated frequency patterns obtained by a sinusoidal function with a frequency in the range [1-5]. We labelled the image as negative if the pattern's frequency is in the ranges [2-2.5] or [4-4.5]. The rest of the frequency patterns are labeled as belonging to the positive class. Then, we randomly rotate and translate the image pattern, add white noise (with std=0.2) and normalize each image independently. From each image, we randomly sample 200 points in the

Table 5: Test set accuracy and binary cross entropy loss.

|          | MLP  | GCN   | GIN   | GAT   | ChebNet |
|----------|------|-------|-------|-------|---------|
| Accuracy | 50   | 77.90 | 87.60 | 85.30 | 98.2    |
| Loss     | 0.69 | 0.454 | 0.273 | 0.324 | 0.062   |

$100 \times 100$ image plane and we divide the image into 200 regions by watershed algorithm (Meyer, 1994), where each sampled point is the marker. From this preprocessing, we generate 5000 graphs, each graph having 200 nodes. Each node corresponds to a watershed region in the image, and if the two regions have intersection on the image plane, we assume these two nodes are connected by an edge in the graph. We set the average intensity value in each region as a 1-length node feature. Even though we know the region center position, we do not use it in order to make the problem harder. Sampled generated image, randomly selected points and their watershed regions, and the graph can be found in Figure 13 for a 30-node illustration.

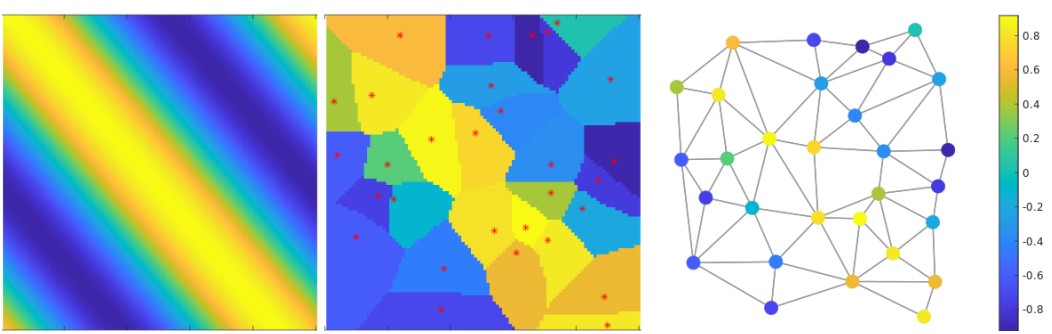

Figure 13: Sample graph in Band-Pass graph dataset. Random rotated and translated image pattern with frequency of 1, random sampled points and their watershed regions, and graph represent the connected region and average region intensity value respectively.

We divided the dataset into train/valid/test subsets, with respectively 3000, 1000 and 1000 graphs. We resampled the same number of positive and negative examples, such that the dataset is balanced. We used 3 layers of GNN followed by a mean readout layer and finally two fully connected layers which have 10 and 1 neuron respectively. Since the problem is a binary graph classification problem, we used binary cross entropy loss and no regularization. We roughly use 30K parameters in each model. The dropout ratio has been applied to all GNN layer's inputs and optimized with respect to the validation set performance.

The results are found on Table 5. Since the node distributions are all the same in the graphs (because the graph nodes were independently normalized), MLP cannot do better than a random classifier. GCN does not perform well, probably because of its low-pass nature. Since GAT and GIN are better than GCN according to the spectral ability, they got a better accuracy than GCN. Finally, ChebNet with 5 convolution supports clearly outperforms the rest of them with a huge margin. These results show that models able to catch a particular band of frequencies obtain the best results, whereas only low-pass based methods like GCN perform only slightly better than MLP. Therefore, this toy example confirms our theoretical analysis.

To conclude, we have shown that if the model is able to perform different filtering operation, it can classify the graphs according to frequency of its signal.

## K  WHY LOW-PASS GNNS GIVE REASONABLE RESULTS ON SEMI SUPERVISED TASKS?

In the recent literature, GNNs are generally evaluated on semi-supervised node classification problems. The most well-known datasets are Cora, CiteSeer and PubMed paper citation graphs (Yang et al., 2016). In these graphs, each node corresponds to a paper. If one paper cites another one,

Table 6: Comparison of methods on the transductive learning problems using publicly defined train, validation and test sets. Results are on accuracy

| Method | Cora | CiteSeer | PubMed |
|---|---|---|---|
| MLP | 0.551 | 0.465 | 0.714 |
| ChebNet | 0.812 | 0.698 | 0.744 |
| CayleyNet | 0.819 | 0.701 | 0.751 |
| GCN | 0.819 | 0.707 | 0.789 |
| GAT | 0.830 | 0.725 | 0.790 |
| **LowPassConv** | $0.827 \pm 0.006$ | $0.717 \pm 0.005$ | $0.794 \pm 0.005$ |

there is an unlabeled and undirected edge between the corresponding nodes. Binary features on the nodes indicate the presence of specific keywords in the corresponding paper. The learning task is to attribute a class to each node (i.e., paper) of the graph using for training the graph itself and a very limited number of labeled nodes. Labeled data ratios are 5.1%, 3.6% and 0.3% for Cora, CiteSeer and PubMed respectively. Since the connected node's probability of being in the same class is high (0.83, 0.71, 0.79 for Cora, CiteSeer and PubMed respectively in Liu et al. (2020)), these graphs are classified as assortative graphs. When the connected nodes are highly likely to be in the same class, label propagation based low-pass effected algorithms can give reasonable results.

To show empirical evidence that any ordinary low-pass filter can give comparable results by low-pass GNNs, we created a fixed, spectral-designed, single convolution support GNN whose frequency response is manually designed in the spectral domain by $\Phi(\boldsymbol{\lambda}) = (\mathbf{1} - \boldsymbol{\lambda}/\lambda_{\max})^5$. This GNN model is denoted as **LowPassConv** in Table 6 and its average accuracy and standard deviation over 20 random runs reported. We use predefined train, validation and test sets as defined by Yang et al. (2016) and follow the test procedure of Kipf & Welling (2017) and Veličković et al. (2018) for a fair comparison. According to Table 6, spectral-designed GNN's such as CayleyNet and ChebNet are slightly outperformed by other methods, including our simple low-pass convolution GNN. On the other hand, GCN and GAT do not give significantly better results than ordinary low-pass graph convolution.

These results seem conflicting with the idea of having spectral well-designed graph convolutions. However, if the problem just needs low-pass filtering effect and if the model produces some unnecessary spectral component in the output, it may have a negative effect on the accuracy. Instead of just one single low-pass convolution support, if we have many convolution supports with different spectral properties, it increases the trainable parameters for a vain. Regularization may help to overcome this issue. This problem may be solved by learning the convolution support in the frequency domain by another secondary unsupervised task which is in our priority list to do.

## L  DATASETS AND IMPLEMENTATION DETAILS

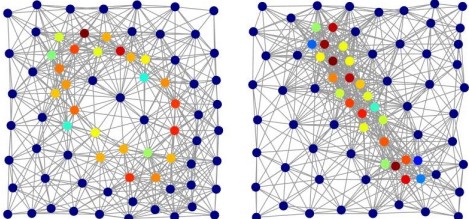

Figure 14: Two sample graphs in MNIST-75 dataset (from 0 and from 1 class), the location of the nodes is just for illustration. Models do not use the node positions.

In our experiments, we used 3 citation graph datasets, named Cora CiteSeer and PubMed, an artificial regular graph where each node degree is 2, named 1D, 2 biological graph datasets named

Table 7: Summary of the datasets used in our experiments.

|  | Cora | CiteSeer | PubMed | 1D | Band-Pass | PROTEINS | ENZYMES | MNIST-75 |
|---|---|---|---|---|---|---|---|---|
| # Graphs | 1 | 1 | 1 | 1 | 5K | 1113 | 600 | 70K |
| # Nodes | 2708 | 3327 | 19717 | 1001 | 200 each | 39.06 each | 32.63 each | 75 each |
| # Edges | 5429 | 4732 | 44338 | 1001 | 1072.6 each | 72.82 each | 62.14 each | 694.7 each |
| # Features | 1433 | 3703 | 500 | NA | 1 | 3 | 21 | 1 |
| # Classes | 7 | 6 | 3 | NA | 2 | 2 | 6 | 10 |
| # Training | 140 Nodes | 120 Nodes | 60 Nodes | NA | 3K | 9-fold | 9-fold | 55K |
| # Validation | 500 Nodes | 500 Nodes | 500 Nodes | NA | 1K | 1-fold | 1-fold | 5K |
| # Test | 1000 Nodes | 1000 Nodes | 1000 Nodes | NA | 1K | NA | NA | 10K |

PROTEINS and ENZYMES, Band-Pass graph dataset which was created by us in order to evaluate the models, and a large scale MNIST-75 graph dataset. The details of these datasets can be found in Table 7. Two samples in MNIST-75 dataset are shown in Figure 14.

In our tests on MNIST-75, all hyperparameters are tuned by a grid as follows: $\ell$-2 norm regularization applied on trainable weights in $\{0, 10^{-1}, 10^{-2}, 10^{-3}, 10^{-4}, 10^{-5}\}$ and dropout in $\{0, 0.2, 0.4, 0.6\}$ for all models. In CayleyNet, we treated the zoom parameter as an hyperparameter and tuned it within the candidate values $\{0.5, 1, 1.5, 2\}$ and also $r \in \{1, 2, 3\}$ which leads $\{3, 5, 7\}$ numbers of supports. ChebNet support number is also another hyperparameter, we tuned it in set of $\{3, 5, 7\}$. For GAT, we tuned the number of heads and the number of output features by concatenating or not which can give the predefined layer feature. For a 64-feature output, two architectures were studied: 8 heads each has 8 outputs with concatenating, and 8 heads each has 64 output without concatenating. The same for 128-feature output layer as well, using the same number of heads. For each layer of GIN model, there is one main GIN convolution layer followed by the same size of MLP. We tested the fixed $\epsilon$ and trainable as well; finally we concluded to use trainable one according to a validation set result. All activation are ReLU, but Elu (exponential linear unit) in GAT. In the output layer, the linear activation is used in all models and the loss function is the cross entropy. We used Adam optimization with a 0.01 learning rate without decay. We fixed the number of iterations to 100 under 64 batch size. The test results were selected on the iteration where the validation set accuracy is maximum.

