# OpenReview forum: "Analyzing the Expressive Power of Graph Neural Networks in a Spectral Perspective"
_ICLR.cc/2021/Conference — ICLR 2021 Poster_

### Official Review · AnonReviewer2 · 2020-10-28
**An interesting perspective for analyzing Graph Neural Network**

**Rating:** 8
**Confidence:** 4

**Review:**

==== Summary ====

This work studies the performance of different Graph Neural Network (GNN) from a spectral perspective. In particular, it shows the kernel for all kinds of proposed GNN models can be expressed in a general form with a specific frequency response definition, which indicates the spectral property (spectrum) of the kernel. Based on this definition, this work empirically studies the band-pass property for kernels in different models and demonstrate the importance of such spectral perspective.

==== Pros ====

+ I like the spectral analysis, which is interesting, novel and appears to be important.
+ The discussion on different GNN models is comprehensive and solid enough.
+ The experimental result shows that it is important to have different kinds of filters

==== Cons ====

I do not have too much criticism about this work. I am interested to see if there could be some extra discussion/example about how should we choose the kernel for some specific problem, based on the spectral perspective. The experimental result in Sec 5.2 shows that for tasks on images using some kernel with spectrum cover the whole region has the best performance, then *how about other tasks such as semi-supervised learning?* It would be great if there is some guidance/analysis, otherwise we still need to determine the most suitable model based on empirical result.

Another question is that, the expressive power analyzed by WL-test involves the depth of the network, while in this work it appears that only one layer is considered, so I am wondering *what is the relation between the depth and the expressive power, from a spectral perspective?* To me it seems to be difficult to analyze, but I believe it is worth to address, as we have seen the importance of depth in CNNs.

==== Reason for score ====

This paper is clear-written, the spectral perspective looks very interesting to me, the theoretical analysis appears to be correct and solid, and the experimental result verifies that the analysis from such perspective is important indeed.  Therefore I tend to vote a clear accept.

==== Minor comments ====

- Below Eq. (2), '$g_0, g_1$ ... trainable models', a typo?

---

> ### Author Response · Authors · 2020-11-21
> **Selection of Convolution Supports, semisupervised settings, and filtering effect when we use deeper network.**
>
> We thank the reviewer for this review and to point out good points and to bring useful interrogations to improve the paper.
>
> Below we bring some detailed replies to the reviewer questions :
>
> Q1a : How to select Convolution Supports
> ---
>
> We agree with the reviewer; we have been asking ourselves the same question of the choice of the kernel for some specific task/dataset. This problem can be seen in a first way as hyperparameter tuning, but further work is needed to be able to learn to solve this task.  While we don’t solve it right now, we think the present paper provides some insights. The general recommendation is to cover the whole spectrum, and not to limit the filters to low-pass. However,  when we train the frequency response in the spectral domain, we have seen that the model easily overfits and cannot generalize the problem. Addressing the selection and/or learning of frequency responses of convolution kernels constitutes one of our top priority future work. One idea we would like to explore is to learn the frequency response by a secondary task in an unsupervised way.
>
>
> Q1b :  semi-supervised settings.
> -------------------
>
> Concerning semi-supervised learning, we have conducted experiments on three well-known assortative (connected node’s probability of being in the same class is high) benchmark datasets  in Appendix K
> These results in Table6, show that low-pass filtering based GNNs (e.g., GCN and GAT) provide good results and even outperform spectral designed ChebNet.
> We didn’t include these results in the body of the paper but in appendix K because it seems to us obvious that assortative transductive learning (i.e., semi-supervised learning) on data such as social networks naturally promotes low-pass filters. Thus, using something other than low-pass filters increases the number of parameters of the model, and thus may decrease the generalization power.
> However, there are some new researches [a,b] on disassortative semi-supervised problems, where the connected node’s probability of being in the same class is lower. This point makes the problem harder for low-pass convolution supports and we strongly think that spectrally well-designed or learned graph convolutions may help the generalization of disassortative semi-supervised problems.
>
>
> [a] Pei, Hongbin, et al. "Geom-gcn: Geometric graph convolutional networks." ICLR (2020).
>
> [b] Liu, Meng, Zhengyang Wang, and Shuiwang Ji. "Non-Local Graph Neural Networks." arXiv preprint arXiv:2005.14612 (2020).
>
>
> Q2 : the relation between the depth and the expressive power
> ---
> We completely agree with this comment. Our spectral analysis shows the single convolution’s effects on the single band of input.
> If there are many hidden layers, each layer applies the given convolution to the previous one’s output. We can see this process like applying the convolutions back to back. After each layer output, there is linear transformation and non linear activation. If we neglect the non-linear activation, the network’s final output will be a weighted sum of many times filtered signals. Thus if the convolution is low-pass, it just applies low pass on to the low-passed signal again and again. So the final output’s frequency profile will be more and more smoothed input signal. However if the GNN have many different spectral effect convolution supports, the final output will be a weighted sum of each signal convolution, but this time there is multiplication, For instance, on one signal the first layer applies low pass, second layer applies high pass, so on so forth. By this multiplication of each convolution support, the final output becomes more productive. Unfortunately, it is difficult to show a theoretical guarantee, but we can measure the ability of the multi-layered GNN by simulation. We will consider this idea for our future work.
>
> Unfortunately, it is difficult to show theoretical guarantees but we can measure the learning ability of multi-layered GNNs by simulation or by some artificial tasks like we propose in Appendix I. In the proposed experiments, we measure the different filtering capabilities of GNN models. To do that, we filter an original realistic image by manually designing low-pass, band-pass and high-pass filters and we measure the learning ability of GNN by each filtered image with respect to going deep in layers or going wide in convolution kernel. We have seen that going deep and wide makes the chebnet more powerful in terms of ability to produce any desired spectral filter.
> Please refer to this appendix I, which also shows that GCN and GAT have no ability to learn something rather than low-pass and that GIN has the ability to learn high pass. Nevertheless, these models provide poor performances with respect to spectral designed convolutions.

---

### Official Review · AnonReviewer4 · 2020-10-28
**+ Paper makes explicit connection between spectral and spatial GNNs and shows frequency response of common GNNs.  - Technical novelty? = Can be improved: Strong conclusion from empirical evaluation,**

**Rating:** 6
**Confidence:** 2

**Review:**

## Summary

This paper shows an equivalence framework between Graph Neural Networks (GNNs) defined in the spatial domain (based on local node neighbourhoods updates) and spectral domain (based on filters defined on eigenvalues of the graph Laplacian). Using this framework,  the paper derives the spectral equivalent of common spatial GNNs hence showing these act  low-pass filters in the spectral domain. The paper shows experimentally that on MNIST superpixel dataset, spatial GNNs show poor results compared to spectral GNNs.

## Recommendation
I recommend this paper be accepted (Accept/weak accept given reviewer confidence - 2/5)

## Strong and weak points
+ The paper does not present a new method but makes clear the connection between spatial-designed and spectral-designed GNNs which has not been previously documented to my knowledge. The connection between spatial and spectral GNNs is remarked upon in Bruna's original 2013 paper and is discussed in later papers (e.g., Kipf&Welling GCN, Wu+2019a_ICML). However, this paper makes the connection explicit.

- On the flip side, the technical novelty in the paper might not be sufficient but I would still argue that the paper merits publication.

- One weak point of the paper is the lack of empirical evaluation on downstream tasks. I understand this is not the main focus of the paper  and these numbers are reported in the respective papers but I would like to see the numbers (even if directly taken from literature). The MNIST superpixel dataset gives a highly biased view for need of high-pass filter (which is provided by spectrally-designed GNNs). For example, see [Wu2019a, ICML'19] Tables or Table 2 in https://arxiv.org/pdf/2003.00982.pdf .


## Questions to the authors
- Since the theoretical frequency response of CayleyNet, ChebNet, GCN, GIN and GAT,  is one of the main contributions of this paper,  I think the paper would benefit from presenting this in a tabular form in the main text e.g.,  Method, Classification (design(spatial/spectral), support (fixed/trainable), frequency response dependent on graph structure, etc.)

- The paper mentions the necessity of high-pass and band-pass filters. However, the paper does not show any toy example of a GNN which acts as a  band-pass filter.

- The paper mentions Weisfeiler-Lehman test in the introduction as a theoretical tool for expressive power of GNNs and its limitations in that for "as powerful as k-WL" GNNs, the test cannot distinguish between their discriminative power. However, k-WL for $k \ge 2$ implies non-local updates since multisets are considered. So is the above framework limited to 1-WL GNNs?

---

> ### Author Response · Authors · 2020-11-21
> **Toy examples and new empirical evaluation**
>
> We thank the reviewer for this review and to point out good points and to bring useful interrogations to improve the paper.
>
> Comment 1-2  new method
> --
> We totally agree that our paper does not present a new GNN and that our aim was rather (i) to theoretically show the connection between spatial-designed and spectral-designed GNN, (ii) to provide a general framework that enables a deep spectral analysis of existing approaches, and (iii) to discuss the results of this analysis. These contributions would be very useful to the community for the design of future GNNs.
>
> Since, our general framework includes most of the recent reference GNNs, by Eq3, many methods can be explained through their Convolution supports. We think that it would be the clearest way to explain GNN models and show the differences. We also think that the new taxonomy can certainly help researchers in this field.
>
> Another important contribution of the paper, is the spectral analysis of existing approaches and its conclusion. Concerning Chebnet, even if the frequency response is already known, it has never been explicitly shown. Concerning Cayleynet’s, its working scheme was given in the spatial domain, but the authors did not show their convolution supports behavior in the spectral domain and the interchangeability between spatial-spectral domains. That is why our theoretical analysis of CayleyNet is original and useful. Concerning GIN, we show how the epsilon and average node degree act on the frequency response, and the spectral analysis of GAT have never been done in the literature.
>
> Comment 3  empirical evaluation
> -------------------
> That is true that the first version of the paper didn't present extensive tests on graph problems to show the advantage of spectrally well-designed filters since it is not the main focus of our study. However, we agree that such results may be interesting to corroborate our claims. Thus, we have added two toy examples and results in appendix I,J and semisupervised problem in appendix K to keep the main paper straightforward but bring this information.
>
> Concerning MNIST, methods in the literature use the 2d positions of the center of the superpixel region, which makes the problem easier because pixel position is a strong inductive bias. This information is generally not available in a realistic graph problem. That is why we exclude node positions. So our results on Mnist superpixel are not comparable with the mentioned tables.
>
> Q1 : Tabular form
> --
> Thanks for this recommendation. The summary table is in the main text now.
>
> Q2 : Toy example
> ---
> We agree with the reviewer that a toy example is a good way to attest our claims. In this way, we integrate the  appendix I and J new experiments which are respectively a node regression problem and a graph classification problem.
>
> In the first experiment in appendix I, we propose an empirical analysis on a real image and its corresponding 2D regular grid graph. We create three different spectral filters that correspond to low-pass, band-pass and high-pass effects and apply these filters to the given input image. We train the GNN models to minimize the square error between its output and targeted filtered image. In this way, we measure the learning capability of the models for band-pass high-pass and band-pass filter. We have seen that GCN, GIN and GAT cannot learn band-pass, and they are not good to learn high-pass, but just low-pass.
>
> With this second toy problem in appendix J, we want to measure the generalization ability of GNN for graph classification problem where graph classes depend on the signal that the graphs carry. We generate 5000 images composed of random generated frequency patterns obtained by sinusoidal function with frequency and labeled according to pattern's frequency.  We randomly divide the image into 200 regions by watershed algorithm and generate the graph each having 200 nodes. Each node corresponds to a watershed region in the image, and if the two regions have an intersection on the image plane, we assume these two nodes are connected by an edge in the graph. Results again show us how GCN, GIN, GAT are not good to generalize the graphs according to its signal's spectral profile.
>
> Q3 :  limited to 1-WL GNNs?
> --
> Yes, we agree with the reviewer that our theoretical spectral analysis is on node-level message passing (1-WL). As long as the algorithm performs node level message passing, even though it is as powerful as higher-order WL like GIN (proved that it is equivalent to 2-WL), our spectral analysis can be directly applied. However, the higher-order WL equivalent GNNs  uses message passing between higher-order structures (C. Morris paper) or non-local updates (H. Marron paper) which our current framework cannot work. This framework can be extended easily to do spectral analysis of GNN for signals on k-order structures.  But current framework can empirically evaluate the spectral ability of all GNN include higher-order WL as in Appendix I

---

### Official Review · AnonReviewer1 · 2020-10-28

**Rating:** 6
**Confidence:** 4

**Review:**

1. Summary:

The authors study the expressive power of GNNs from the spectral view and bridge the gap between spectral and spatial designs of graph convolutions by casting several popular GNN models into a common framework. The authors also shows the necessity of non-low-pass filters on some datasets.

2. Strong points of the paper:

a) The writing is quite clear to understand.

b) The experiments indicate the need to go beyond low-pass filtering.

3. Clearly state your recommendation (accept or reject) with one or two key reasons for this choice:

I am slightly leaning towards recommending a rejection to this paper, with the main reason being that I am not sure if the contributions are sufficiently significant, as I will elaborate below.

4. Provide supporting arguments for your recommendation:

a) Regarding the results on the frequency responses obtained for the different models, those on ChebNet and CayleyNet seem not surprising since the spectrum is their starting point. The filtering properties of GCN have also been relatively well-studied, such as in [1, 2, 3, 4, 5], plus equation (10) only holds when the graph is roughly regular.

b) In section 5, some of the claims ask for more justifications, such as “GCNs need a high number of convolutional kernels if the input-output pairs can be figured out certain simple filters”, and “since GIN is not spectral-designed, there is no guaranty that it works the same for different graph datasets”. They could be correct in the end, but it will be better to support them with either theoretical arguments or experimental evidence. For example, could you come up with specific examples that show the limits of GCNs and GINs mentioned here?

c) For GAT, it is not clear whether using randomly sampled attention weights is representative of its normal filtering behavior.

5. Ask questions you would like answered by the authors to help you clarify your understanding of the paper and provide the additional evidence you need to be confident in your assessment:

a) In what way is the theoretical analysis here helpful for our understanding of the different kinds of GNNs beyond existing work on the filtering properties of GNNs?

b) As I mentioned above, it would be nice if some of the claims can be further supported.

c) On Page 3, the second line, what exactly does “$\lambda_i$” mean? Does “i” refer to the i-th feature dimension or the i-th eigenvector?

6. Additional feedbacks:

Possible typo: On Page 6, the third line from the bottom, should it be Figure 2a and Figure 2b instead of Figure 1a and Figure 1b?

References:

[1] Felix Wu, Tianyi Zhang, Amauri Holanda de Souza Jr, Christopher Fifty, Tao Yu, and Kilian Q Weinberger. Simplifying graph convolutional networks.

[2] Hoang NT and Takanori Maehara. Revisiting graph neural networks: All we have is low-pass filters.

[3] Kenta Oono and Taiji Suzuki. Graph neural networks exponentially lose expressive power for node classification.

[4] Qimai Li, Zhichao Han, and Xiao-Ming Wu.  Deeper insights into graph convolutional networks for semi-supervised learning.

[5] Jiezhong Qiu, Yuxiao Dong, Hao Ma, Jian Li, Kuansan Wang, Jie Tang. Network Embedding as Matrix Factorization: Unifying DeepWalk, LINE, PTE, and node2vec.

---

> ### Author Response · Authors · 2020-11-21
> **First part: Clarify the contribution, why our gcn's frequency response is important**
>
> We thank the reviewer for the time spent to review our paper and the constructive comments and valuable feedback.
> To clarify our contribution, we provide generic connections, based on theoretical foundations, between spectral and spatial designed convolutions. Based on a theorem that we establish, we analyze many well-known graph neural networks, including original analyses as argued below.
> We agree that some of our claims were not completely supported in the first version of the paper. That is why we have added a set of experiments in the new version to fill this hole. We hope the obtained results will convince the reviewer.
>
> Comment 4a and Q1
> ---
> In addition to the spectral analysis, our general framework explains most of the GNNs by (Eq3). We strongly think that it would be the clearest way to explain different GNN models and show the differences between the methods, especially from an educational point of view. We also think that the general framework and the new proposed taxonomy can certainly help newcomers to this field. Using the proposed framework, many methods can be explained through their Convolution supports.
>
> Concerning Chebnet, the reviewer is right, the frequency response is already known. However, it has never been explicitly shown.
> Concerning Cayleynet’s, its working scheme was given in the spatial domain, but the authors did not show how their convolution supports work in the spectral domain and the interchangeability between spatial-spectral domains. That is why our theoretical analysis of CayleyNet is original and useful.
> Concerning GIN, we show how the epsilon and average node degree act on the frequency response, which has never been done in the literature.
> Finally, the spectral analysis of GAT’s frequency response has never been done in the literature as well.
>
> We agree that given references [1, 2, 3, 4, 5]  present studies that have some common objectives with ours. However,  they are mainly restricted to the study of Kipf’s GCN while our study includes more models.
> Moreover, our analysis differs from these existing studies in an important aspect.
> The mentioned papers study the filtering properties of GCN by considering the identity matrix added to the adjacency matrix to form the convolution kernel as part of the structural information of the graph.
> As an example, in [1], the frequency response of GCN is obtained using what the authors call the augmented normalized Laplacian  (see section 3.2 line 17 in [1]) which is created by adding self node connections adjacency. The paper argues that the corresponding GCN’s frequency response has the same shape as a GCN’s initial convolution support (without “renormalization trick”)  but "renormalisation trick" just shrinks the eigenvalues, as shown in Figure 2 rightest plot of the paper[1].
> We agree that from a practitioner's point of view, both have almost the same meaning. However, either the given graph adjacency has or has not self added connection, we strongly think that frequency response should be with respect to the original laplacien but not augmented one.  Thus we think that our definition of GCN’s frequency response is more realistic and accurate.
>
> The reviewer is right that Eq10 only theoretically holds if the graph is roughly regular. However, we do not think that this point is a weakness of our theory but on the contrary an important point. Hence, Eq10 is an approximation because the GCN is not spectral-designed as it is classified in the literature. Eq10 shows that GCN frequency response is not exactly the same for each graph. It also shows how its frequency response changes with respect to the average node degree. Mentioned spectral analysis on GCN did not see that point. We are hoping to highlight this spectral misclassification of GCN through this paper.
> Moreover, in Fig2a, Fig5 and Fig6, we computed the frequency response for many diverse graphs and saw that the general shape is almost the same.
>
> Comment 4b) Related Q2
> --
> multi convolution support:
> --
>
> Kipf’s GCN does not need a high number of convolution kernels, since it is designed by a single low pass filter. However, ChebNet and CayleyNet could benefit from multiple convolutional kernels to create bandpass filters, as for GAT using multiple convolution heads. The more convolution kernels we have, the more accuracy we have in the band associated with each filter. This statement is corroborated by new experiments provided in appendix J in table 4 for ChebNet
>
> GIN has no guarantee that it works the same for different in spectral domain
> --
> Spectral-designed graph convolutions should always have the same frequency response. Empirically, as illustrated in Figures 2(b) and 2(c), GIN does not provide the exact same profiles in the spectral domain. Its frequency response changes according to the given graph’s average node degree and also its trainable epsilon parameters. The epsilon effect is illustrated in figure Fig2b-c. That fact is also proved in Theorem5.

---

> ### Author Response · Authors · 2020-11-21
> **Second Part: Toy examples, Comment4c and typos**
>
>
> Specific examples that show the limits of GCNs and GINs
> ---
>
> In order to illustrate these limits, we propose in appendix I and J new experiments on two toy problems which are respectively a node regression problem and a graph classification problem.
>
> Node regression problem :
> In the first experiment (please see appendix I), we propose an empirical analysis on a real image with resolution of 100$\times$100 and its corresponding 2D regular 4-neighborhood grid graph.  We create three different spectral filters that correspond to low-pass, band-pass and high-pass effects and apply these filters to the given input image.  We train the GNN models to minimize the square error between its output and targeted filtered image.
>
> In Table 3, the results show that ChebNet constantly outperformed GCN, GIN and GAT for all tasks. For learning low-pass filtering, the rest of the models did better compared to the high-pass and band-pass tasks. That is the fact that GCN, GIN and GAT have the ability to act as low-pass filters. In addition to doing better on the low-pass task, GIN also did relatively better on the high-pass task as well. It is obvious that GIN can work as a high pass if the $\epsilon$ parameter is selected negative (see Theorem5). It turns out that the trained values of $\epsilon$ in GIN for each layer are $-5.27$, $-2.21$ and $-0.47$ for the high-pass task.  Please see appendix I for further details.
>
> Graph classification problem:
> With this second toy problem (please see appendix J), we want to measure the generalization ability of GNN for graph classification problem where graph classes depend on the signal that the graphs carry. We generate 5000 images of size 100x100 composed of random generated frequency patterns obtained by sinusoidal function with frequency in range of [1-5]. We labelled the image as negative if the pattern’s frequency is in range of [2-2.5] or [4-4.5]. Rest of the frequency patterns are assumed as belonging to the positive class. Then, we randomly rotate and translate the image pattern, add white noise with std=0.2 and normalize each image independently. From these images, we randomly sample 200 points in the 100x100 image plane (for each image) and we divide the image into 200 regions by watershed algorithm where each sampled point is the marker.
> From this pre-processing, we generate 5000 graphs, each graph having 200 nodes. Each node corresponds to a watershed region in the image, and if the two regions have intersection on the image plane, we assume these two nodes are connected by an edge in the graph. We set the average intensity value in each region as a 1-length node feature.
>
> The results given in Table 5 show that MLP cannot do better than random classifier. GCN does not perform well, because of its low-pass effect according to our analysis. Since GAT and GIN are better than GCN according to spectral ability, they got a better accuracy than GCN. Finally, Chebnet with 5 convolution supports clearly outperforms the rest of them with a huge margin. These results show that models able to catch a particular band of frequencies obtain the best results, whereas only low pass based methods like GCN performs only slightly better than MLP. Therefore, this toy example confirms our theoretical analysis.
>
>
> Comment 4c) Related Q2
> --
> Why randomly generated attention weight is useful
> ---
> Since there is no way to calculate its theoretical frequency response to the best of our knowledge, we would like to show its limits on a sample graph named Cora. During the training, what the model learns will be in the same distribution as our simulations. However, we also showed the learned attention weights' frequency response in fig3b-c for Enzymes and Protein graph dataset. In the analogy with the euclidean convolution 3x3 mask, since the weights during training cannot be known in advance, we can do simulation by producing random weights in 3x3 mask and we can calculate the frequency response of these masks. In that simulation, we can see that 3x3 convolution mask frequency response covers the spectrum. Thus we can conclude that 3x3 euclidean mask has the ability to produce many different kinds of frequency profile output. We did the same to measure GAT's frequency ability.
>
>
>  what exactly does “$\lambda_i$” mean?
> -------------------------------
> There $\lambda_i$ stands for the i-th eigenvector. We understand that it is confusing since we overuse ‘i’. We modify the paper by using the index k and s for node indexes and filter indexes, to be clearer.
>
>
> Typos
> -----
> Indeed there were some mistakes in figure references. It is fixed.

---

### Official Review · AnonReviewer3 · 2020-10-30
**R3**

**Rating:** 8
**Confidence:** 4

**Review:**

In this paper, the authors propose a spectral-based analysis method to analyze the modeling abilities of major GNNs. Specifically, the first use the concept of convolution support to unite the ideas of spatial-based methods and spectral-based methods. By further identifying the frequency profile of different models, the authors obtain an overview of which spectrum range different models may cover. The evaluation on regular and in-regular graph datasets validate their arguments. In general, the paper brings an interesting perspective in addition to the WL-test to reveal the expressive power of GNNs, and both the theory and evaluation sound solid. It would be better if the authors can further address a few issues.

First, the authors summarize that in certain cases (e.g., sum pooling), the spatial based methods can be considered in the form of Eqn. 3 shows. Though the authors describe that advanced methods like GAT can also be described in this way, it would be great if the authors can make an explicit argument in the beginning of Sec. 3 to describe whether general spatial based methods (e.g., using max pooling, or other $upd(x, y)$ functions) can also be described in this way.

I also suggest the authors to revise Definition 2 to further explain why each convolution support has the same frequency response over different graphs in a spectral designed case, as there might be some training parameters that can be affected by different graph structures. In addition, in Corollary 1.1, the authors indicate the frequency profile of any given graph convolution support can be defined as Eqn. 7, it is unclear if this also includes spatial-designed graph convolution.

The analysis in Sec. 5 is interesting, but it would be better if the authors can expand the discussion on how we can further utilize the obtained frequency profile to analyze the expressive power of different GNNs. Dividing different filters into low-pass, high-pass, and mid-band categories seem to be in a very coarse granularity and it would be better if the authors can discuss some potential directions for more detailed analysis. I’m also interested in if the number of kernels of the same category (e.g., low-pass filters) may affect model performance. It would be interesting to see if any experimental results are provided to analyze the impact of the number of kernels, the distribution of types rather than their categories only.

In summary, this work provides a new perspective in analyzing the expressive power of GNNs and I suggest the authors to further address the above issues.

---

> ### Author Response · Authors · 2020-11-21
> **Different number of convolution supports and layers effect to the frequency profile**
>
> We thank the reviewer for this review and to point out good points and to bring useful interrogations to improve the paper. He raises many questions which are at the heart of our current and future works !
>
> Below we bring some detailed replies to the reviewer questions :
>
> Q1 : General framework includes anisotropic GNN
> --
> Yes, it would be good to start section 3 with the claim that the proposed framework covers many GNN includes anisotropic graph convolution. However, we just exclude the one who uses max or std_dev aggregation operators, for which there is no theoretical or strong empirical evidence that they improve the accuracy. A sentence has been added at the beginning of section 3 in the new version.
>
> Q2 revise Definition 2 and Corollary 1.1,
> --
>
> We define a spectral convolution as a convolution which performs in the same way for any kind of graph in the spectral domain. Even if the convolution support is not fixed (has some trainable parameter), that parameters should be the same for all graphs and the frequency response does not change from one graph to another. For trainable support, yes sure the frequency response changes during the training, but it changes as the same way for different graphs in spectral point of view. This point is the weak point of spectral designed convolutions, because it is not directly affected by spatial differences of the graphs. But for sure spatial differences effects eigenvectors, thus the convolution is affected undirectly.
> Concerning Corollary1.1, yes it is valid for all graph convolutions whose support was given by a C matrix in our framework, including spatial-designed ones. We modified our sentence to make this aspect clearer in the new version.
>
> Q3 : utilization of the obtained frequency profile.
> --
> Indeed our current expressive analysis cannot be represented by any numerical measure, but just visual. The frequency response of the convolution support is the figure that represents its expressiveness. In order to discuss potential direction, we can say its passing and stopping spectral regions, if it covers the spectrum or not, and if it has any band specific filters or not. Moreover we can do simulations and we can show the productive limits of GNN model even if it has multi layers. Addressing this aspect constitutes one of our top priority future work
>
> Q4:  number of kernels
> --
> If a GNN has many different spectral effect convolution supports on a single layer, (and if we neglect the non-linear activation effect) the final output will be a weighted sum of each signal convolution. If the GNN has multi layers, convolution effects are multiplied. By this multiplication of each convolution support, the final output becomes more productive. If the supports are all different kinds of low pass convolutions, again the output becomes a weighted sum of these effects, but since each has similar effect, the output will not be rich in terms of spectral view.
>
> We have illustrated the impact of the number of kernels on a toy problem in appendix I. We propose an empirical analysis on a real image with resolution of 100$\times$100 and its corresponding 2D regular 4-neighborhood grid graph.  We create three different spectral filters that correspond to low-pass, band-pass and high-pass effects and apply these filters to the given input image.  The problem can be viewed as a single graph node regression problem, where we train the GNN models to minimize the square error between its output and targeted filtered image.
>
> In order to assess ChebNet, GCN, GIN and GAT we use a 3-layer GNN architecture whose input is a one-length feature (intensity of the pixel) and the number of neurons in hidden layers is respectively 32, 64 and 64; the output layer is an MLP that projects the final node representation onto the single output for each node.
>
> In Table 3, the results show that ChebNet constantly outperformed GCN, GIN and GAT for all tasks. For learning low-pass filtering, the rest of the models did better compared to the high-pass and band-pass tasks. That is the fact that GCN, GIN and GAT have the ability to act as low-pass filters. In addition to doing better on the low-pass task, GIN also did relatively better on the high-pass task as well. It is obvious that GIN can work as high pass if the $\epsilon$ parameter is selected negative (see Theorem~5). It turns out that the trained values of $\epsilon$ in GIN for each layer are $-5.27$, $-2.21$ and $-0.47$ for the high-pass task.
>
> Table 4 studies the ability of ChebNet with respect to the number of convolution supports and number of layers. It shows that the given frequency response becomes better when the number of convolution supports and number of layers increases. However this result is not surprising where it is proved that any frequency response can be written by a weighted sum of enough number of Chebyshev polynomials.

---

### Author Response · Authors · 2020-11-21
**New version of the paper includes toy examples and discussion on semisupervised setting problems**

After receiving very fruitful questions/recommendations from reviewers, we submitted a new version of the paper. Mainly our new version includes 3 new appendices. These are;

Appendix I:  Which Filters can the GNN Models Learn?
--
We propose an empirical analysis on a real image and its corresponding 2D regular grid graph. We create three different spectral filters that correspond to low-pass, band-pass and high-pass effects and apply these filters to the given input image. We train the GNN models to minimize the square error between its output and targeted filtered image. In this way, we measure the learning capability of the models for band-pass high-pass and band-pass filter. We have seen that GCN, GIN and GAT cannot learn band-pass, and they are not good to learn high-pass, but just low-pass.

Appendix J:  Can GNN Classify Graphs According to Frequency of its Signal?
--
We want to measure the generalization ability of GNN for graph classification problem where graph classes depend on the signal that the graphs carry. We generate 5000 images composed of random generated frequency patterns obtained by sinusoidal function with frequency and labeled according to pattern's frequency. We randomly divide the image into 200 regions by watershed algorithm and generate the graph each having 200 nodes. Each node corresponds to a watershed region in the image, and if the two regions have an intersection on the image plane, we assume these two nodes are connected by an edge in the graph. Results again show us how GCN, GIN and GAT are not good to generalize the graphs according to its signal's spectral profile.

Appendix K:  Why low-pass GNNs give reasonable results on Semi supervised Tasks?
--
Concerning semi-supervised learning, we have conducted experiments on three well-known assortative (connected node’s probability of being in the same class is high) benchmark datasets.  These results show that low-pass filtering based GNNs (e.g., GCN and GAT) provide good results and even outperform spectral designed one. On the other hand, any ordinary manually designed single convolution support low-pass filter can give comparable results by low-pass GNNs.

Typos
--
Also, we fixed some typos and added a table to summary the studied methods as the reviewers pointed out.

---

### Comment · ~Muhammet_Balcilar1 · 2021-02-07
**Codes and Datasets**

We released the codes and datasets.
They are available via:
https://github.com/balcilar/gnn-spectral-expressive-power

---

### Decision · Program_Chairs · 2021-01-07
**Final Decision**

**Decision:**

Accept (Poster)

**Comment:**

This paper gives a new theoretical tool to connect the gap between the spectral perspective and spatial perspective of graph neural networks. The frame-work is considerably broad and can deal with several existing methods. From this view point, the connection between the spatial and spectral perspectives are made explicit while they are noticed in an informal way by existing researches. The frequency response of several methods are analyzed through theories with support by some numerical experiments.

The idea of connecting spatial and spectral perspective would not be entirely new, but the main novelty of this paper is to make it explicit and analyzed the frequency response of well-known methods concretely. This is informative to the literature and extends some known results to more general settings. The numerical experiments well justify the plausibility of the theory. For reasons mentioned above, I think this paper is worth publishing in ICLR2021.